

# Glacier geometry limits the propagation of thinning in Patagonian Icefields

Bastian Morales[1], Marcelo Somos-Valenzuela[2], Mario Lillo[1], Iñigo Irarrazaval[3], David Farias[4], Elizabet Lizama[1], Diego Rivera[5], Alfonso Fernández[4]

[1]Faculty of Agricultural Engineering, Universidad de Concepción, Av. Vicente Mendez 595, Chillan, Chile
[2]Department of Forest Sciences, Faculty of Agriculture and Environmental Sciences, University of La Frontera, Temuco 4780000, Chile
[3]Centro de Investigación en Ecosistemas de la Patagonia, Coyhaique, Chile
[4]Departamento de Geografía, Universidad de Concepción, Victor Lamas 1290, Concepción, Chile
[5]Facultad de Ingeniería, Universidad del Desarrollo, Avenida Plaza 600, Santiago, Chile

*Correspondence to*: Marcelo Somos-Valenzuela (marcelo.somos@ufrontera.cl)

**Abstract.** Climate change is causing a decline in glaciers globally, with the possibility that some may disappear during this century. Recent findings postulate that the geometric glacier-topography configuration has the capacity to limit glacier thinning upstream. The Patagonian Icefields (PI), with 15,900 km² of glaciers, are the world's largest glacial freshwater reservoir after Antarctica and Greenland. In recent decades, it has been one of the areas with the greatest mass loss worldwide due to climate change. Our research explores the relationship between glacier geometry and changes in PI glaciers to determine regions vulnerable to thinning. We studied 45 major marine- and lake-terminating glaciers in PI using the Péclet number (Pe) based on the diffusive kinematic wave model to determine the geometric state of glaciers and as a metric of vulnerability to diffusive thinning. Locations with Pe ≤ 8 experienced greater thinning and retreat, suggesting an empirical limit that encompasses more than 90% of ice thinning. The empirical limit is related to a significant change in the gradient of slope and roughness of the subglacial topography at PI due to a knickpoint in the subglacial bed. On average, ~53% of the total ice flow of PI glaciers is below the thinning limit. Therefore, due to the current geometric state and evolution, lake-terminating glaciers may propagate frontal thinning deep inland. The empirical thinning limit provides signals of priority glaciers to investigate considering current climate change projections.

## 1 Introduction

Due to anthropogenic climate change, glaciers are shrinking globally. Glaciers in mountainous regions have suffered a loss of $123 \pm 24$ Gt yr$^{-1}$ during 2006-2015 (IPCC, 2022). This widespread loss of glacier mass allows





us to project that, depending on global warming trajectories, some glaciers could disappear entirely in the world's mountain ranges during the 21$^{st}$ century (IPCC, 2021; Huss et al., 2017). Moreover, South American glaciers have
contributed significantly to sea level rise (SLR) as mass balance estimates indicate that glaciers have lost 22.9 ± 5.9 Gt yr$^{-1}$ annually since 2000. The most significant mass loss is concentrated in the Patagonian Andes, totaling 21.9 ± 5.8 Gt yr$^{-1}$ (Dussaillant et al., 2019). This value is 50% greater than the SLR contribution of all high mountain glaciers in Asia (Brun et al., 2017), although the latter cover an area three times larger (Dussaillant et al., 2019).

The role of glacier geometry in the evolution of glaciers is crucial to reassessing the trajectories of large glaciers in the future and their contribution on a global scale (Felikson et al., 2017; Zheng et al., 2019; Felikson et al., 2021; Zheng, 2022). Glacier geometry, which refers to the shape and size of a glacier resulting from its interaction with topography, plays a crucial role in ice flow dynamics (Felikson et al., 2017; Zheng et al., 2019; Felikson et al., 2021; Zheng, 2022), in the access of meltwater and consequent basal lubrication (Zheng, 2022;
Yan et al., 2021; Riel et al., 2021; Armstrong et al., 2022; Joughin et al., 2013), in the balance of forces (Catania et al., 2020; Pfeffer, 2007; Van Der Veen, 1996; Enderlin et al., 2018) and in the evolutionary response of the glacier to a perturbation in the terminus (Felikson et al., 2017; Felikson et al., 2021; Zheng, 2022). When a glacier retreats, the ice at the terminus experiences complex stresses, including longitudinal, transverse, and shear stresses. These stresses are influenced by various factors, including the glacier bed's slope, the glacier terminus's shape,
and the ice's properties (Carnahan et al., 2022; Bondzio et al., 2017).

The geometric control on glacial evolution as a consequence of topographic variability is well described in tidewater glaciers west of Greenland, where terminus position observations over 30 years have been compared using satellite remote sensing and subglacial topography data (Catania et al., 2018a). In this region, it has been found that glacial retreat accelerates through wide and overdeepenings parts of the bed, which is characterized by
retrograde slopes. Also, overdeepenings regions of deep incision can be used as a predictive measure of retreat duration, while that short bed regions with prograde bed slopes are insufficient to stop terminus retreat (Catania et al., 2020, 2018a). The authors postulate that regions with bed overdeepenings may drive glacial retreat, because they concentrate subglacial runoff which carries water into fjords. Therefore, glacial geometry exerts relative importance on different processes that influence the detachment of the terminus, such as buoyancy in the terminal region, buttressing provided by ice mélange, and scour through melting caused by water body-glacier contact
(Catania et al., 2020).



The geometric state of ice flow has the potential to define regions of upstream stability so that, depending on the location of these limits, the degree of vulnerability of a glacier to thinning could be deduced (Felikson et al., 2017, 2021; Zheng, 2022). These discoveries, based on the model of diffusive kinematic waves (Nye, 1963, 1960; Weertman and Birchfield, 1983), have used the Péclet number as a metric to define these regions (Felikson et al., 2017, 2021). The Péclet number corresponds to the relationship between advection and diffusion of the kinematic wave equation in glaciers ( Felikson et al., 2017; Zheng et al., 2019; Felikson et al., 2021; Zheng, 2022). Kinematic wave theory establishes that glacial thinning can be modeled as a wave propagating upstream due to a perturbation in the terminus. In this sense, Carnahan et al. (2022), based on the study of Greenland glaciers, suggest that regions with low Péclet numbers substantially impact the components of the force balance, highlighting that glaciers with large areas of low basal resistance extending inland from the terminus allow a chain of stress changes that results in sustained acceleration, more significant mass loss, and continued retreat.

Despite significant advances, gaps remain in our knowledge about how glacial geometry limits changes in Patagonian Icefields  and, if possible, through these conditions to define regions of vulnerability to thinning. The above is particularly important because the Patagonian Icefields (46.2-51.5°S) is a territory covered by approximately 15,900 km² of glaciers, is one of the most significant ice masses in the world and the main temperate ice mass in the southern hemisphere (Aniya et al., 1996), with significant potential to affect global sea level (IPCC, 2021). In addition, it corresponds to the main water reserve of glacial origin in the Andes. In recent decades, this region has been one of the areas with the most significant mass loss due to climate change, contributing approximately a 3 mm increase in sea level between 1961 and 2016 (Minowa et al., 2021). In this region, we still do not understand how glacier geometry contributes to the propagation of terminal changes.  We hypothesize that the geometry of the ice flow limits the propagation of thinning in the Patagonian Icefields, controlling its evolution over time. This research seeks to fill knowledge gaps by 1) analyzing the relationship between glacial geometry and changes in glaciers of Patagonian Icefields. 2) determining through the geometric state of the ice flow regions vulnerable to thinning.

To carry out this research, we used the Péclet number based on the diffusive kinematic wave model as a glacier vulnerability metric, widely used in Greenland glaciers (Felikson et al., 2017; Zheng et al., 2019; Felikson et al., 2021; Zheng, 2022). We analyzed the relationship between this geometric metric and the thinning, front changes, and force balance of the Patagonian Icefields glaciers over the last two decades. Finally, we  evaluate the role of bed topographic variability in the changes in the main glaciers of the Andes.



## 2 Methodology

### 2.1 Study area

The Patagonian Icefields, located in South America between latitudes 46.2° and 51.5°S, are composed of the Northern (NPI) and Southern (SPI) Icefields, covering an area of approximately 15,900 km$^2$. NPI is located between 46.2° and 47.5°S and covers a total ice area of 3,953 km$^2$ (Rivera et al., 2007). It has a north-south elongation of approximately 100 km and a width of 40-45 km (Aniya, 1988; Carrasco-Escaff et al., 2023). NPI shows steep topography with terrain elevation values increasing eastward over most of the ice field area, reaching sea level at the western margin and a maximum of 3,970 m a.s.l. at the summit of Mount San Valentin. Characteristic terrain elevation values are 1,000 m a.s.l. for the west side and 1,500 m a.s.l. for the east side (Warren and Sugden, 1993; Carrasco-Escaff et al., 2023). NPI comprises 38 major outlet glaciers (Dussaillant et al., 2018), including the San Rafael Glacier, being the main marine-terminating glacier covering 18% of the NPI area. There are 18 lake-terminating glaciers covering 64% of the surface, and 19 land-terminating glaciers covering 18% of the NPI surface (Rivera et al., 2007; Willis et al., 2012; Pfeffer et al., 2014; Collao-Barrios et al., 2018). The SPI is located between 48.33° and 51.5°S, covering a total ice area of 12,514 km$^2$ (Casassa et al., 2014). It extends approximately 350 km long and is generally between 30 and 40 km wide, with a narrow part approximately 8 km wide (Aniya et al., 1998). SPI contains a central plateau that extends between 1,400 and 2,000 m a.s.l., with terrain elevation values decreasing to the south (Carrasco-Escaff et al., 2023). SPI reaches its topographic maximum at Lautaro volcano with a peak of 3,607 m asl. This ice field is composed of 48 main outlet glaciers (Aniya et al., 1996, 1998; Casassa et al., 2014); of these, 16 are marine-terminating glaciers covering 40.09% of the area, 30 lake-terminating glaciers reaching 58.39% of the area and two land-terminating glaciers covering 1.51% of the SPI area (Casassa et al., 2014).

NPI and SPI have glacial valleys with ice thicknesses above 1,400 m and sectors below sea level in the western branch of the Pío XI Glacier- Occidental Glacier, between San Rafael and Colonia glaciers, and near Monte Fitz Roy (Millan et al., 2019). Ice volume estimations from thickness data is 4,756 ± 923 km$^3$ (Millan et al., 2019). Regarding PI dynamics, ice velocity estimates from 1984 and 2014 reveal regions of fast flow extend into the plateau and accumulation area in both ice fields (Mouginot and Rignot, 2015). The quickest glacier in the NPI, the San Rafael Glacier, flows at 7.6 km yr$^{-1}$. The second largest glacier, San Quintin Glacier, flows at 1.1 km yr$^{-1}$ (Mouginot and Rignot, 2015). At SPI, velocities of up to 10.3 km yr$^{-1}$ have been measured for Penguin Glacier,



8.8 km yr$^{-1}$ for Europa Glacier, 6.0 km yr$^{-1}$ for HPS-19, and 5.0 km yr$^{-1}$ for HPS-28 (Mouginot and Rignot, 2015).

During this period, Pio XI Glacier, the largest glacier in the PI, showed a maximum velocity of 2.5 km yr$^{-1}$.

The area changes reported for three decades show a glacier area loss at NPI of -5.0 ± 2.8% between 1986-2005 and -3.5 ± 3.0% during 2005-2016 (Meier et al., 2018). For the same periods, SPI reduced -4.3 ± 2.1% and -2.0 ± 2.3%, respectively (Meier et al., 2018). On the other hand, frontal changes reported between 2000-2019 (Minowa et al., 2021) reveal retreats of up to -132 m yr$^{-1}$ (Steffen glacier) at NPI and -291 m yr$^{-1}$ (HPS 12 glacier)

at SPI. In turn, advances of 2 m yr$^{-1}$ (Leones glacier) have developed at NPI (Minowa et al., 2021). Meanwhile, on the Pio XI glacier at SPI, advances of up to 50 m yr$^{-1}$ have been detected in the northern outlet and 32 m yr$^{-1}$ in the southern outlet (Minowa et al., 2021). A geodetic mass balance between 1976/1979 and 2000 shows a reduction of -0.63 ± 0.03 m w.e. yr$^{-1}$ for 63% of NPI and -0.33 ± 0.05 m w.e. yr$^{-1}$ for 52% of the glaciated areas of SPI (McDonnell et al., 2022). For the same areas analyzed between 1976/1979 and 2000, between 2000-2020,

the geodetic mass balance was -0.78 ± 0.03 m w.e. yr$^{-1}$ at NPI and -0.80 ± 0.04 m w.e. yr$^{-1}$ at SPI (McDonnell et al., 2022). Between 2000-2012/2015, Braun et al. (2019) estimated a geodetic mass balance of -0.85 ± 0.07 m w.e. yr$^{-1}$ for NPI and - 0.86 ± 0.08 m w.e. yr$^{-1}$ for SPI. For their part, Dussaillant et al. (2019) calculated between 2000-2015 a geodetic mass balance of -0.88 ± 0.26 m w.e. yr$^{-1}$ for NPI and -0.90 ± 0.29 m w.e. yr$^{-1}$ for SPI. Investigations have revealed upward balances regarding PI's modeled surface mass balance (Schaefer et al., 2013,

2015; Carrasco-Escaff et al., 2023). The simulation periods for these studies have been concentrated between approximately 1975 and 2015.

Climate processes that modulate weather conditions in Patagonia have shown significant changes in recent decades due to global climate change (Garreaud, 2018; Cai et al., 2020). The climate of the Patagonian Icefields is influenced by the topographic configuration, westerly winds (Garreaud et al., 2009), the Southern Annular

Mode (Thompson and Wallace, 2000; Garreaud et al., 2013), and El Niño Southern Oscillation (ENSO) (Bravo et al., 2019; Gómez et al., 2022). The westerly flow transports large amounts of moisture from the southern Pacific Ocean to the continent. Air parcels are lifted by convection over the Andes, leading to orographic precipitation and, consequently, high precipitation with strong W-E gradients (Carrasco et al., 2002; Garreaud et al., 2013; Collao-Barrios et al., 2018). The climate of PI is temperate and very humid, with precipitation exceeding 10,000

mm in the highest zone (Garreaud et al., 2013). The equilibrium line altitude (ELA) is located at 950-1,300 m a.s.l. at NPI (Rivera et al., 2007; Collao-Barrios et al., 2018). At SPI, ELA estimates are 800-900 m a.s.l. for the western slope and 1,500-1,600 m a.s.l. for the eastern slope (De Angelis et al., 2007). The proposed ratio of solid precipitation to total precipitation is 0.596 in PI (Schaefer et al., 2015), but recent findings show that this ratio



may vary locally (Bravo et al., 2019). In PI, solid precipitation shows higher rates on the west than east, with mean

differences ranging from 1,500 mm w.e to 3,500 mm w.e (Bravo et al., 2019). Analyses of data from weather

stations located on the eastern and western sides of the ice fields and reanalysis data at specific grid points speak

to the PI's climate change signal, showing warming trends from 0.3°C to 1.4°C (Schaefer et al., 2013). Data from

the El Calafate weather station 50°30'S) near the Moreno glacier, in SPI, show a warming trend of 0.3°C in the

period 1940-1990 (0.006°C yr$^{-1}$) (Schaefer et al., 2013; Ibarzabal y Donangelo et al., 1996). A warming of 0.5°C

at 850 hPa between 1960 and 1999 (0.013°C yr$^{-1}$) was found by analyzing reanalysis data at grid point 50°S, 75°W

(Rasmussen et al., 2007; Schaefer et al., 2013). Analyses of 30 years of land temperature data showed a weak

trend (not significant at 90% confidence) between 38°S and 48°S (Falvey and Garreaud, 2009). In the region,

significant interannual and interdecadal variations in precipitation occurs (Schaefer et al., 2013), and regional

precipitation trends are spatially non-homogeneous (Aravena & Luckman, 2009; Carrasco-Escaff et al., 2023; R.

Garreaud et al., 2013; Quintana & Aceituno, 2012). Negative trends have been reported towards the north of NPI

(Boisier et al., 2018) and positive trends towards the south of SPI (González-Reyes et al., 2017).

In our research, 45 marine- and lake-terminating glaciers were studied in NPI and SPI, with areas greater than

or equal to 65.5 km$^2$. This study covered more than 80% of the total area occupied by glaciers in the region,

representing approximately 12,000 km² of glacier surface (Fig. 1). Next, the data used in the research, the

theoretical bases of the methods used, and the workflow developed for data analysis are described.



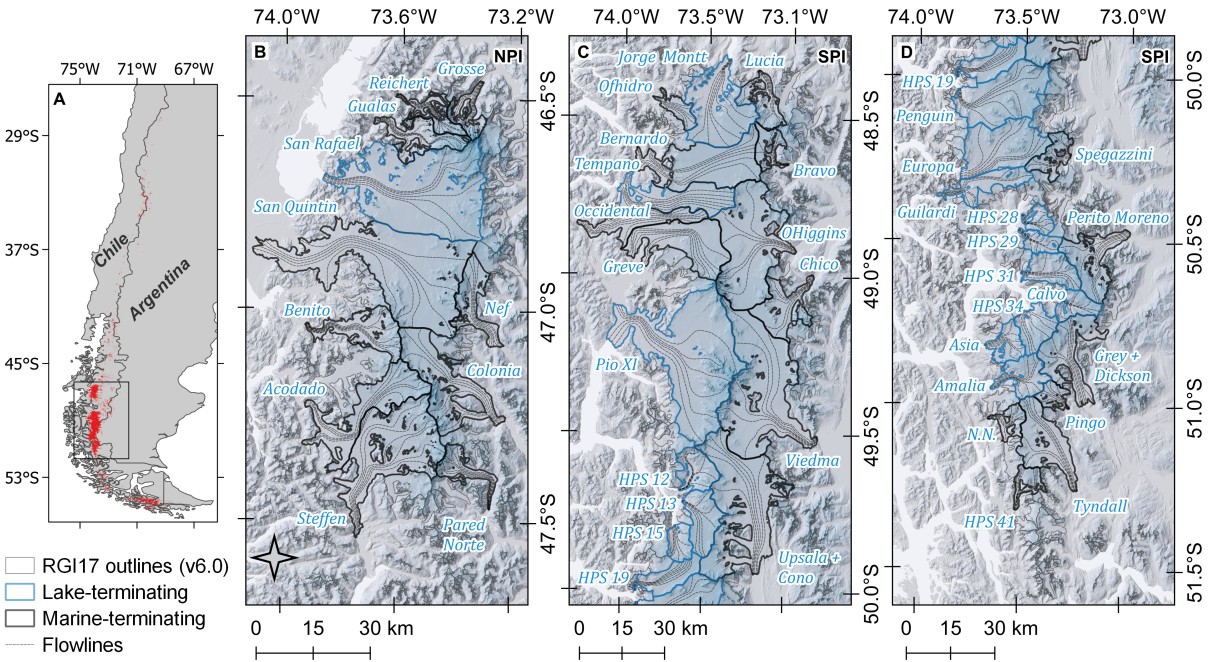

**Figure 1.** Study area. Patagonian Icefields. Marine- and lake-terminating glaciers are delineated next to the flowlines for each PI glacier. A) Spatial context of PI in RGI17 v6.0. B) NPI glaciers. C) SPI glaciers. D) Continuation of SPI glaciers.

## 2.2 Data sources

We collected multiple geospatial and glaciological data for one-dimensional temporal analysis of changes in PI between 2000 and 2018. The baseline glaciological data used in our research corresponds to the Randolph Glacier Inventory (RGI) v6.0 glacier inventory contour of the year 2000 for PI and the classification of glaciers according to the type of term. We used the digital elevation model (DEM), SRTM V3, provided by NASA with 30 m spatial resolution, to represent the terrain elevation in 2000. For 2018, we represented the terrain elevation through the elevation difference between the SRTM DEM and the total elevation change between 2000-2018 of Dussaillant et al. (2019). The exclusive use of the product of Dussaillant et al. (2019) is based on the need to achieve temporal coherence of the data to develop the analysis. The subglacial topography was estimated through the difference of the SRTM V3 DEM and the global coverage glacier thickness model developed by Farinotti et al. (2019). This





model, with a resolution of 50 meters, uses up to five different models to provide a consensus estimate of the ice thickness distribution. It is based on the principles of ice flow dynamics to determine thickness from surface characteristics. The subglacial topography model generated for PI has a spatial resolution of 50 m. Glacier surface velocity for the year 2000 was provided by the Inter-Mission Time Series of Land Ice Velocity and Elevation (ITS_LIVE) project, part of NASA's Making Earth System Data Records for Use in Research Environments

(MEaSURES) 2017 program, which provides global low-latency measurements of glacier surface velocity (Gardner et al., 2019). The ITS_LIVE data product is a set of regional compilations of annual mean surface velocities for major glacier-covered regions from 1985 to 2018 (Gardner et al., 2019). To estimate the velocity for the year 2000, we used the average velocity between 1999 and 2000 derived from Landsat 4, 5, and 7 images and processed through the auto-RIFT method described in Gardner et al. (2018). The period 1999-2000 is defined

as stationary to reduce the presence of null pixels by 59.68%, compared to the velocity product of the year 2000. For 2018, we used the velocity product of Millán et al. (2022) of 50 m resolution, developed using radar satellite data interferometry. Null values for elevation, velocity, and thickness were filled by interpolation using a four-way conical search and inverse distance weighting over three neighboring pixels. We used the Fill-nodata tool from the GDAL (Geospatial Data Abstraction Library) geospatial data translation library in QGIS 3.22.

For the one-dimensional analysis of PI glaciers, we manually delineated six main flowlines for each glacier using the gradient of the previously described velocity products as a mask. Through this approach, we seek to ensure appropriate spatial routing of glacier flowlines. After manual delineation, the flowlines were set a fixed spacing of 50 m through the vector analysis tools of QGIS 3.22.

## 2.3 Peclet number

Along glacier flow, thinning perturbations that are initiated by terminus retreat evolve as a diffusive-kinematic wave, which moves downstream with the flow and diffuses both upstream and downstream (Nye, 1960; Felikson et al., 2021). The dimensionless relationship between the advection coefficient ($C_0 - \frac{\partial D_0}{\partial x}$), and the diffusion coefficient $D_0$, of the diffusive-kinematic wave is expressed by the Péclet number ($Pe$) (Felikson et al., 2017; Zheng et al., 2019; Felikson et al., 2021; Zheng, 2022). In these coefficients, $C_0$ corresponds to $\frac{\partial q}{\partial H}$ and $D_0$ to $\frac{\partial q}{\partial \alpha}$.

Where $q$ is the ice flow (with a dimension of $L^2/T$), $H$ corresponds to the ice thickness and $\alpha$ is the slope of the glacier. For its part, $x$ represents a position along the flowline and $\partial x$ the distance changes along it (positive from the terminus) (Zheng et al., 2019; Felikson et al., 2017). The advection coefficient determines the velocity of the



kinematic wave (the motion of the perturbation as a whole), while the diffusion coefficient determines the strength of diffusion (how fast a perturbation can propagate spatially) (Zheng et al., 2019). The relationship between the two coefficients is expressed as follows:

$$Pe = \frac{(C_0 - \frac{\partial D_0}{\partial x})}{D_0} l \tag{1}$$

Where $l$ is the length of a perturbation. If $Pe$ is much greater than 0, forward advection will dominate, and any ice thickness perturbation will only propagate downstream (Zheng, 2022). This prevents destabilization in the upper current if glacier thinning or retreat begins near the terminus. On the other hand, if $Pe$ has a value less than or equal to 0, upward diffusion or advection occurs, and a thickness perturbation at the terminus can propagate upstream, changing the dynamics of the entire glacier (Zheng, 2022). Therefore, we considered a glacier with low $Pe$ to be more vulnerable than one with high $Pe$ (Felikson et al., 2017; 2021; Zheng, 2022).

Following Zheng (2022), $Pe$ can be rewritten as follows:

$$Pe = [\frac{(m+1)\alpha_0}{mH_0} - \frac{\partial U_0}{\partial x}\frac{1}{U_0} - \frac{\partial H_0}{\partial x}\frac{1}{H_0} - \frac{\partial \alpha_0}{\partial x}\frac{1}{\alpha_0}]l \tag{2}$$

Where $m$ corresponds to the hard-bed sliding law parameter (Weertman, 1957), whose value is set to 3, $\alpha_0$ is the slope of the glacier, $H_0$ is the glacial thickness, $U_0$ is the glacial speed, and $l$ corresponds to the perturbation length.

We used the workflow Zheng (2022) to analyze the Péclet number for Patagonian Icefields glaciers in one dimension. Using the outlined flowlines, we extracted an average Pe based on the six flowlines outlined for each glacier, considering a spacing of 50 m along the flowline. We excluded from the analysis the first 250 m of the terminal region (5 vertices) to reduce edge effects associated with the influence of velocity values outside the RGI v6.0 limits. We applied the Savitzky–Golay filter to reduce data noise following the methodology of Zheng (2022). Subsequently, we estimated the average Pe values obtained through the flowlines to create a vector data set with the average Pe for each of the PI glaciers evaluated for subsequent analysis. We used the NumPy, Pandas, and SciPy libraries for data processing in a Python 3.11 programming environment.





## 2.4 Analysis of Glacier Changes

### 2.4.1 Elevation and Front Changes

This study used two models to analyze elevation changes (geodetic) in PI glaciers. The first model, developed by Dussaillant et al. (2019), is based on the differences between the Shuttle Radar Topography Mission (SRTM)

Digital Elevation Models (DEM) and Advanced Spaceborne Thermal Emission and Reflection Radiometer (ASTER) optical satellite products providing a temporal range of analysis from 2000 to 2018. In complement, the model of Braun et al. (2019) was used, which was developed through spatiotemporal differences between the SRTM DEM and TanDEM-X, covering the period from 2000 to 2016. We extracted the average elevation changes in distance units along the flowline for the 45 glaciers considering the six flow lines outlined for each glacier. We

applied the Savitzky-Golay filter to reduce data noise following the methodology of Zheng (2022). We used the NumPy, Pandas, and SciPy libraries for data processing in a Python 3.11 programming environment.

To calculate front changes, contours from the 2000 Randolph Glacier Inventory (RGI) v6.0 glacier inventory (Pfeffer et al., 2014) and the multitemporal glacier inventory for the Patagonian Andes developed by Meier et al. (2018) up to 2016. For this last inventory, the authors delineated the glaciers through a semi-automated process

using satellite images from the Landsat and Sentinel constellation. We estimated front changes in distance units along the center flowline for the 45 glaciers via QGIS 3.22.

Elevation and terminus changes were analyzed with respect to Péclet number, force balance, bed slope, and terminus type.

### 2.4.2 Changes in force balance

To study the changes in the force balance between 2000 and 2018 in PI glaciers, we use the elevation and velocity data described in section 2.2 to calculate the stress fields. In our research, we use the force balance method according to van der Veen & Whillans (1989), which provides us with the stress balance in glaciers for a given instant. The force balance method has been used extensively in Greenland (Carnahan et al., 2022) to understand

glacier behavior, either individually (Van Der Veen et al., 2011) or on multiple glaciers for given periods (Bartholomaus et al., 2016; Stearns and van der Veen, 2018; Van Der Veen et al., 2011; Enderlin et al., 2018;



Meierbachtol et al., 2016). This method has also been applied to glaciers in Antarctica (Price et al., 2002; Stearns et al., 2005) and Alaska (O'Neel et al., 2005; Enderlin et al., 2018).

The force balance method assumes that the driving stress ($\tau_d$) is supported by basal drag ($\tau_b$), depth-integrated

longitudinal coupling ($F_{long}$), and lateral drag ($F_{lat}$). In this work, like Carnahan et al. (2022), we used a sign convention where positive driving stress values act in the direction of flow; in turn, positive values of the other stresses represent opposition to flow:

$$\tau_b = \tau_d - F_{lat} - F_{long} \tag{3}$$

The depth-integrated components in the x-direction are given by:

$$F_{lat} = \frac{\partial}{\partial y}(HR_{xy}) \tag{4}$$

$$F_{long} = \frac{\partial}{\partial x}(HR_{xx}) \tag{5}$$

In this expression, H is the ice thickness and the resistive stresses (R), omitting vertical shear, are given by:

$$R_{xx} = B\dot{\varepsilon}_e^{\frac{1}{n}-1}(2\dot{\varepsilon}_{xx} + \dot{\varepsilon}_{yy}) \tag{6}$$

$$R_{yy} = B\dot{\varepsilon}_e^{\frac{1}{n}-1}(\dot{\varepsilon}_{xx} + 2\dot{\varepsilon}_{yy}) \tag{7}$$

$$R_{xy} = B\dot{\varepsilon}_e^{\frac{1}{n}-1}\dot{\varepsilon}_{xy} \tag{8}$$

Where B is the viscosity rate factor, n is the stress exponent in Glen's flow law, $\dot{\varepsilon}_{ij}$ is the depth-averaged strain rate, and the effective strain rate is given by:

$$\dot{\varepsilon}_e = (\dot{\varepsilon}_{xx}^2 + \dot{\varepsilon}_{yy}^2 + \dot{\varepsilon}_{xx}\dot{\varepsilon}_{yy} + \dot{\varepsilon}_{xy}^2)^{1/2} \tag{9}$$

Here, we assumed that ice flow does not vary with depth; that is, we ignored vertical shear, so the depth-

averaged strain rate is equal to the surface strain rate (shallow-shelf approximation with basal drag) (Carnahan et al., 2022). In PI, it has been suggested that 98% of the surface velocities are due to basal slip (Collao-Barrios et al., 2018). Depth-integrated resistive stresses are estimated with a stress exponent of n = 3 and a viscosity rate factor of B = 300 kPa yr$^{1/3}$, similar to that used in previous investigations (Carnahan et al., 2022; Van Der Veen et al., 2011). The driving stress in the x-direction takes the form:





$\qquad \tau_d = \rho g H \frac{\partial h}{\partial x}$ (10)

Where $\rho$ is the ice density (917 kg/m3), $g$ is the force of gravity (9.81 m/s2), and $h$ is the glacier surface elevation. The basal drag is calculated as the residual of the depth-integrated and driving stresses. For stress processing, we used the workflow developed by Carnahan et al. (2022); for this, we used the NumPy, Pandas, and SciPy libraries in a Python 3.11 programming environment. Subsequently, we performed data extraction along the flowline for the 45 PI glaciers considering the average of the six flow lines outlined for each glacier. We applied the Savitzky-Golay filter to reduce data noise following the methodology of Zheng (2022).

We generated a dataset to analyze using two approaches: the relationship between force balance changes between 2000-2018 (net, percentage, and stress ratio), topography, terminal changes, and Péclet number. In both cases, we classified the Péclet number into eleven classes, from zero to ten. The extremes (zero and ten), include Pe values less than or equal to zero and greater than or equal to ten, respectively. The first data set focused on the behavior of the median force balance along the flow with respect to the thinning limit based on the Péclet number (see section 3.1). The second dataset was developed to analyze the behavior of the force balance in the first 5 km in the terminal region with respect to Péclet number, subglacial topography, and terminal changes (see sections 2.6 and 3.2). We focused on the percentage variations, net and systematic, for a deeper analysis and better representation of the data on the logarithmic relationship of the basal, lateral, and longitudinal stress with respect to the driving stress. For data processing in both analysis approaches and the study of the relationship between force balance, topography, front changes, Péclet number, and type of terminus, a statistical significance and Pearson correlation analysis was performed using the NumPy, Pandas, and SciPy libraries in a Python 3.11 programming environment.


## 2.5 Search for empirical thinning limit

We compared the Pe values with the accumulated percentage thinning to find and delimit the empirical thinning limit. In the case of total elevation change, we calculate the cumulative percentage thinning since the terminus for each glacier across the flowlines. The thinning at position $x$ along the flowline is expressed as a percentage of total thinning as follows:



$$dH(x) = (\frac{-dh(x)}{\sum_{i=1}^{N} -dh(i \cdot 50)}) \cdot 100 \qquad (11)$$

Where $-dh(x)$ is the mean thinning measured in meters at position $x$ along the flowline, while $i$, is the index used to iterate through each position along the glacier flowline, from position 1 to position $N$, with a spacing of 50 meters. Then, to identify the empirical thinning limit, we generated a curve of cumulative thinning versus the
Péclet number for the 45 glaciers. The cumulative thinning from the glacier front in percent is expressed as:

$$CT(x) = \sum_{i=1}^{x/50} dH(i \cdot 50) \qquad (12)$$

Where $x/50$ corresponds to the node position along the flowline considering a 50 m spacing. Subsequently, we divided the Péclet number values into eleven classes and calculated the median thinning for each class in a generalized manner for PI. The Péclet number classes were generated between zero and ten, including at the two
extremes (zero and ten), Pe values less than or equal to zero and greater than or equal to ten, respectively. We established a Pe with a minimum median of 90% of the cumulative PI thinning as an empirical thinning limit through an approach as in Felikson et al. (2017). This empirical thinning limit is expressed as:

$$CT(Pe_{limit}) \geq 90\% \cdot M(Pe_{limit}) \qquad (13)$$

In this equation, $M(Pe_{limit})$ is the median cumulative thinning for the Péclet class $Pe_{limit}$, which represents
the empirical thinning limit. Then, we searched for the average position across the flowline from the terminus (limit position). We estimated the distance to the terminus in each evaluated glacier as follows:

$$CT(Pos_{limit}) = Pe_{limit} \qquad (14)$$

Where $Pos_{limit}$ is the first position from the front of the glacier where it is observed that $CT(Pos_{limit})$ is equal to $Pe_{limit}$. Subsequently, we calculated the percent ice flow under the empirical thinning
limit in PI as:

$$P_{ice\ flow} = \frac{\sum_{i=1}^{Pe_{limit}} CT(i \cdot 50)}{N} \cdot 100 \qquad (15)$$

In this equation, $P_{ice\ flow}$ is the percentage of ice flow considering the $N$ positions along the flowline up to $Pe_{limit}$. We excluded glaciers without a thinning limit and cases where the retreat has exceeded the limit.



## 2.6 Relationship between topography and glacial geometry

Our study explores the connection between bed topography and Péclet number. We considered four bed metrics: elevation, slope gradient, and terrain roughness index (TRI) gradient, which quantifies the variation in terrain elevation by the difference between a central pixel and the surrounding cells (Riley et al., 1999). Lastly, a percentage classification of prograded and retrograded slopes (bed slope type). We obtained the elevation from the SRTM digital elevation model.

Using QGIS 3.22 geospatial processing algorithms, we estimated bed topography utilizing the difference between the SRTM digital elevation model and the glacial thickness model of Farinotti et al. (2019). With this geospatial information, we estimated the bed slope and TRI. We used the bed slope to calculate the slope difference and classify prograded and retrograded slopes. This analysis defined negative slope differences as retrograded and positive slopes as prograded. We calculated the value of the topographic variables from the terminus as a starting point for each of the glaciers, considering the manually delimited flowlines (average of the six flow lines outlined for each glacier). We applied the Savitzky-Golay filter to reduce data noise following the methodology of Zheng (2022). We used the NumPy, SciPy, and Pandas libraries in a Python 3.11 programming environment for data processing along the flowlines.

We generated a dataset to analyze the relationship between topography and Péclet number using two approaches. In both cases, we classified the Péclet number into eleven classes, from zero to ten. In the two extremes (zero and ten), the values of Pe less or equal to zero and greater or equal to ten, respectively. The first dataset generated focused on the topographic characterization of the thinning limit based on the Péclet number (see section 3.1). For this, we concentrated on median elevation, slope gradient, and TRI gradient with respect to the classified Péclet number. The second dataset was developed to perform a generalized analysis of terminus on PI glaciers, including terminus type. For this, we analyzed the slope classification for the first 5 km of the terminus region with respect to the classified Péclet number (see section 3.2) . In both approaches, to study the relationship between topography and Péclet number, a statistical significance and Pearson correlation analysis was performed using the NumPy, SciPy, and Pandas libraries in a Python 3.11 programming environment.



# 3 Results

## 3.1 Glacial Geometry Limits Thinning Propagation

We used the Péclet number based on the diffusive kinematic wave model as a metric of glacial vulnerability to thinning. This analysis evaluated the vulnerability to diffusive thinning of PI's 45 major marine- and lake-terminating glaciers. Data analysis shows that regions with Pe ≤ 8 contain more than 90 % of the cumulative thinning in PI (Fig. 2).

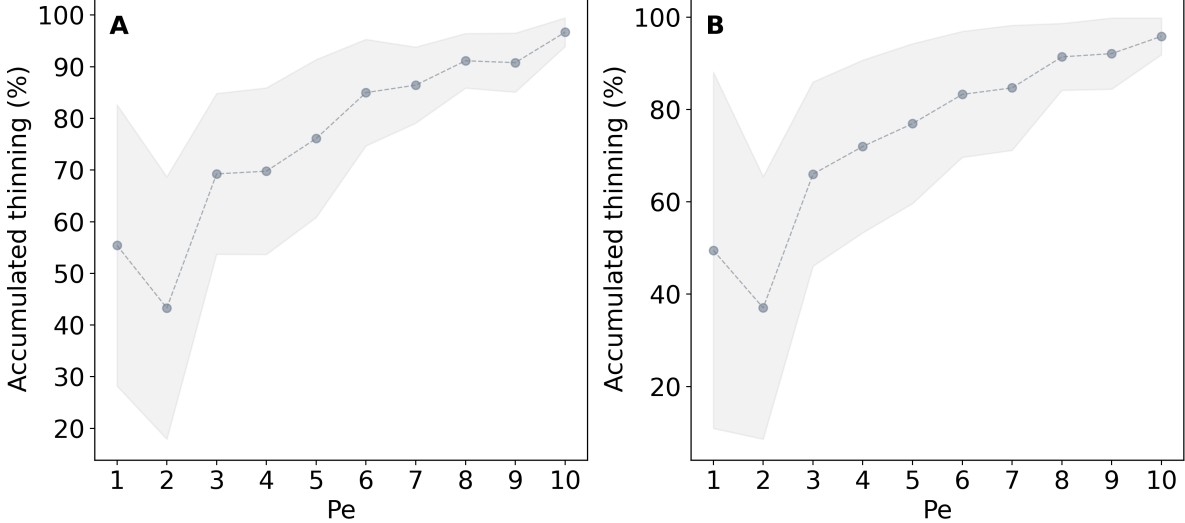

**Figure 2.** The thinning limit is based on the classified Péclet number (average) and the computed median cumulative thinning (median of averages) along ice flow on 45 PI glaciers. A) Thinning limit based on the elevation change model of Braun et al. (2019). B) Thinning limit based on the elevation change model of Dussaillant et al. (2019). The shaded region represents the dispersion of the data estimated through the median absolute deviation.

There is a positive relationship between Péclet number, altitude, and thinning (Fig. 3). This correlation suggests that in areas with altitude lower than 1,104 m a.s.l. (as a median), mass loss occurs in PI by potential diffusion (Fig. 3). However, we observe a critical geometric control of subglacial topography on total elevation





changes in PI. As the value of Pe increases, so does the roughness and slope. These findings suggest that bed roughness and slope have significant implications for the response of glaciers to diffusive thinning. According to our analysis, in the region of the thinning limit (Pe = 8), an abrupt change (drop) in the gradient of slope and roughness is generated in PI (Fig. 3) due to a knickpoint on the subglacial bed. Therefore, given these results, we 360 believe that the morphological characteristics of the bed strongly influence the thinning limit.

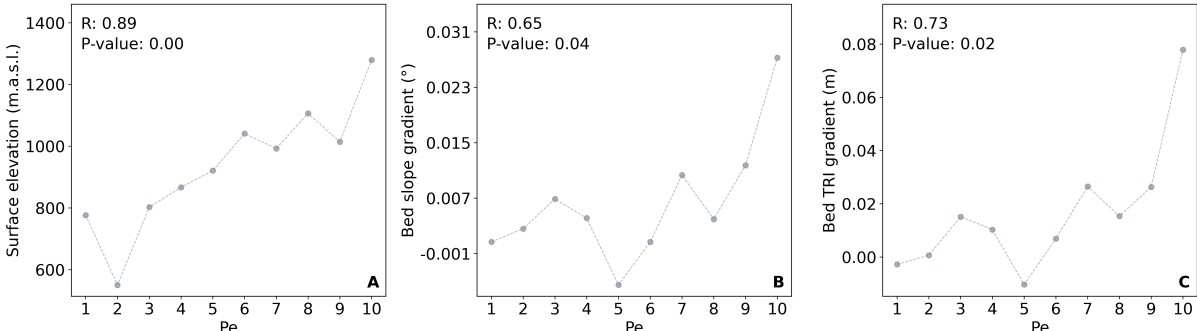

**Figure 3.** Classified Péclet number (average) versus topographic metrics (median of averages) computed along the ice flow in 45 PI glaciers. A) Péclet number versus median elevation. B) Péclet number versus median bed slope gradient. C) Péclet 365 number versus bed median TRI gradient.

The force balance analysis shows substantial changes in basal, driving, lateral, and longitudinal stress (Fig. 4). The data analyzed indicate that the main drops in the force balance are generated under the empirical limit; in this sense, we observed a decline of more than 12% in the basal stress, 26% in the lateral drag and 22% in the 370 longitudinal coupling. On the other hand, the driving stress does not present negative variations. Our data analysis shows that the most significant force balance increases occur in three components at Pe = 7 before the empirical limit. In this sense, basal drag increases by 14%, lateral stress by 29%, and longitudinal coupling by 17%. In turn, the driving stress reaches the highest percentage increase at Pe = 1, with 34%. The increase in driving stress is steadily generated in regions with Pe ≤ 5. Over the empirical limit, we observe a generalized percentage drop in 375 all components of the force balance except for driving stress. According to our results, changes in the force balance components are deeply linked to the geometric characteristics of the subglacial topography. These conditions seem to modulate the propagation of stress changes along the ice flow.



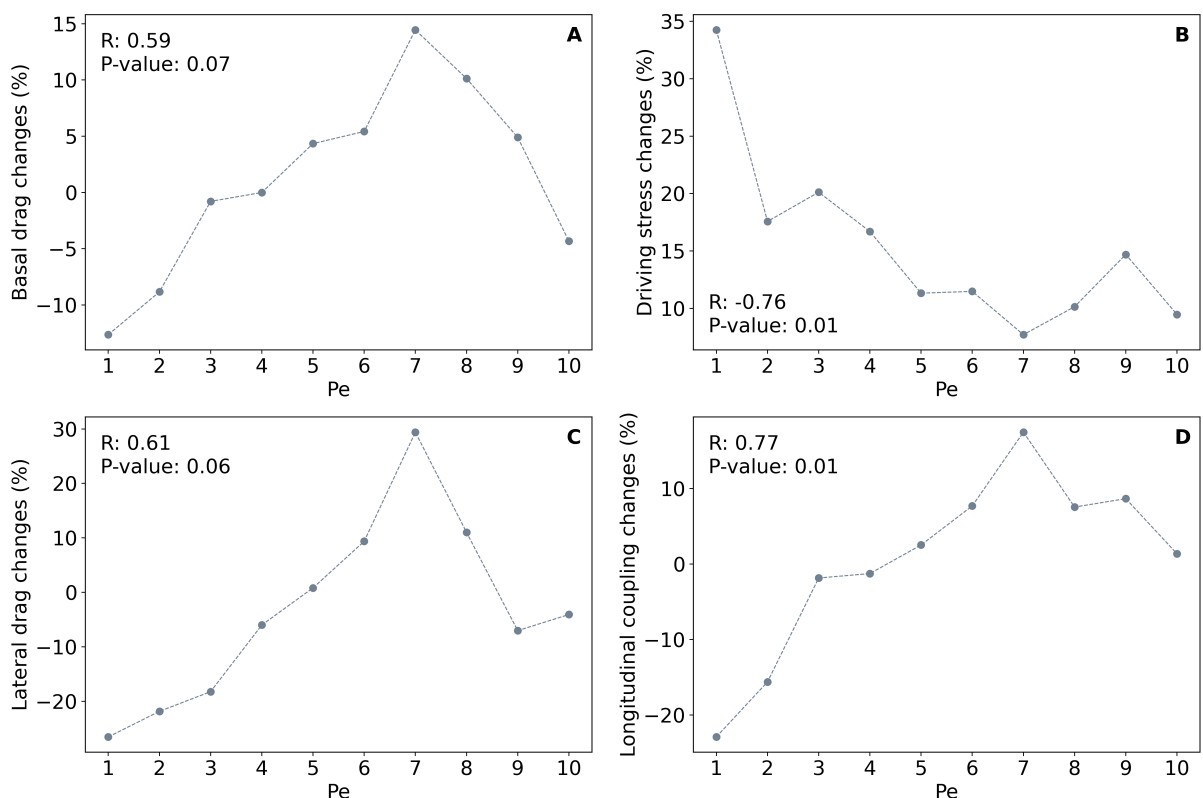

**Figure 4.** The classified Péclet number (average) and its relationship to changes in force balance (median of averages) between 2000 and 2018 are computed along the ice flow at 45 PI glaciers. A) Péclet number versus changes in basal drag. B) Péclet number versus changes in driving stress. C) Péclet number versus changes in lateral drag. D) Péclet number versus changes in longitudinal coupling.

The empirical limit of thinning in PI glaciers indicates that on average ~ 53% of the total flow of PI glaciers is below the thinning limit. Therefore, they are susceptible regions to continue losing mass in the future due to the absence of geometric conditions that prevent the propagation of thinning (Fig. 5, Fig. S1). When comparing the documented retreat in PI and the empirical limit, we note two cases where retreat has exceeded the limit, i.e., ~ 4% of the glaciers analyzed: the Gualas (NPI) and Lucia (SPI) glaciers, both with marine termination. Additionally, it was not possible to find the empirical limit for ~ 9% of the glaciers because they presented a Pe





lower than the established limit along the flow; therefore, all the ice flow is under the empirical limit. The associated glaciers are in SPI and correspond to the marine-terminating HPS 19, Europa glaciers, and the lake-terminating O'Higgins and Greve glaciers. Approximately 87% of the glaciers, the terminus retreat has not yet exceeded the geometric limit. A noteworthy example is the HPS 12 glacier, which retreated ~ 8 km, becoming the
glacier with the highest PI retreat. In this case, the 2017-2018 thinning limit was 3.5 km from the terminus.

We identified glaciers with the highest percentage of ice flow below the empirical limit at PI. At NPI, the San Rafael Glacier, the only marine-terminating glacier, has 73.56% of ice flow below the empirical limit and is located 32.84 km inland. In SPI, we identified an ice flow for the Tempano, Asia, and Pio XI glaciers under the empirical limit of 90.68%, 74.86%, and 72.78%, respectively. Correlatively, for these glaciers, the limit is located
at 40.90 km, 10.90 km, and 41.21 km. At NPI, the glaciers that terminate at the lake and are noted for high ice flow below the empirical limit are the Pared Norte, Benito, and Steffen glaciers. The ice flow percentages for these glaciers are 77.31%, 71.72%, and 67.52%, respectively. In addition, these glaciers' limits are 18.00 km, 20.37 km, and 29.27 km, respectively. On the other hand, in the SPI region, the Bernardo, Pingo, and Chico glaciers present an ice flow below the empirical limit of 82.30%, 77.96%, and 77.01%, respectively. For these
glaciers, the limit is located at 38.78 km, 8.70 km and 17.85 km, correlatively.







**Figure 5.** A) Relationship between thinning limit position and ice flux below the thinning limit during 2000. B) Relationship between empirical limit position during 2000 and terminal changes. C) Relationship between ice flux under the thinning limit



during 2000 and terminal changes. D) Relationship between ice flow under the thinning limit during 2000 and 2018. E) Ice
flow under the thinning limit during 2018 on PI glaciers. Marine-terminating glaciers are represented as a circle, and lake-
terminating glaciers are described as an inverted triangle.

## 3.2 Evolution of the terminus controlled by glacial geometry

We analyzed the influence of geometric control on the evolution of the terminal region: the first 5 km of this
region between 2000 and 2018. The exploratory analysis shows that more than 93% of the Patagonian glaciers
present $Pe \leq 8$. Therefore, practically all the glaciers evaluated show geometric conditions that favor the diffusivity
of upstream thinning. In the terminal region of NPI glaciers, Pe values fluctuate between 1.07 and 2.31, showing
less dispersion with respect to SPI, whose values vary between 0.75 and 11.25 (Fig. 8).

Regarding the behavior of Pe according to the type of terminus, we observed in the marine-terminating of NPI,
San Rafael glacier, a Pe of 1.78. In SPI, the marine-terminating glaciers, Pio XI, HPS 12, and Guilardi, stand out
for a low Pe, with values of 1.04, 1.26 and 1.37, respectively. In lake-terminating glaciers, the Grosse, Gualas,
and Colonia glaciers stand out in NPI, with values of 1.07, 1.23, and 1.28, respectively. In SPI, the Perito Moreno,
Upsala + Cono, and Occidental glaciers have values of 0.75, 0.85, and 1.03. For the cases where no empirical
limit was found, we identified in the marine-terminating glaciers, HPS 19 and Europa, a Pe of 3.98 and 1.61,
respectively. On the other hand, the values of O'Higgins and Greve glaciers, lake-terminating, are 2.17 and 1.89,
respectively.

According to our findings, the greatest retreat-thinning is generated in glaciers with Pe < 4.85 (Fig. 8). In this
sense; we find that low Pe values are related to positions of the empirical limit far inland (from the terminus). For
example, glaciers with Pe values < 4.85 present 59% of ice flux below the empirical limit, while glaciers with Pe
> 4.85 present a remarkable reduction of ice flux below the empirical limit, reaching approximately 22% by 2018
(Fig. 5). Our data analysis reveals that marine-terminating glaciers present a mean Pe value of 3.80 while lake-
terminating glaciers present a value of 1.98. Additionally, marine-terminating glaciers show a limit position closer
to the terminal region than lake-terminating glaciers, with distances of 12.77 km and 18.03 km, respectively (Fig.
S2). This suggests that lake-terminating glaciers may retreat deep into land due to their ability to diffuse thinning
upstream. Thus, the vulnerabilityof these glaciers is manifested in their retreat over the last two decades, with a
mean of -1.76 km for lake-terminating glaciers and -0.77 km for marine-terminating glaciers (Fig. S2).



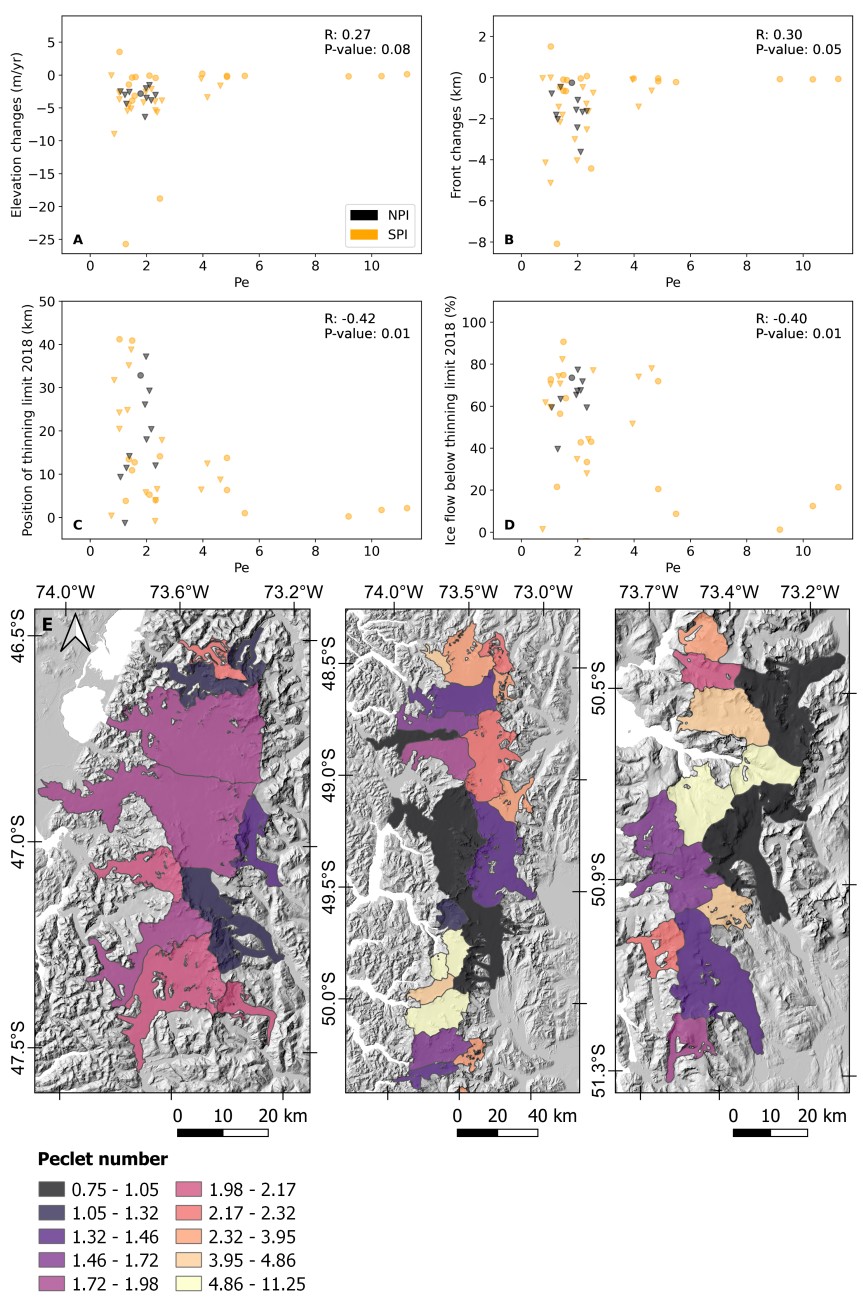

**Figure 6.** A) Elevation changes with respect to the Péclet number (median of averages) in the first 5 km of the terminus. B) Terminal changes with respect to the Péclet number (median of averages) in the first 5 km of the term. C) Position of the



thinning limit in 2018 with respect to the Péclet number (median of averages) in the first 5 km of the terminus. D) Ice flux under the thinning limit in 2018 with respect to the Péclet number in the first 5 km of the terminus. E) Spatial distribution of Péclet number on PI glaciers for the first 5 km of the terminus. Marine-terminating glaciers are represented as a circle, and lake-terminating glaciers are described as an inverted triangle.

We observed a positive linear correlation with Péclet number in three of the four components of the force
balance during the period 2000-2018, highlighting basal, lateral, and longitudinal stress (Fig. S3, Fig. S4). Driving stress shows a slight negative correlation, but without associated statistical significance (Fig. S3, Fig. S4). Significant changes in lateral, longitudinal and basal stresses are visualized in the PI glaciers. According to the analysis, practically all glaciers have evolved to a new state in the components of the force balance because of the changes of the last two decades. In SPI, we observed a substantial increase in basal, lateral, and longitudinal stress
in glaciers with Pe > 3 and in driving stress at Pe < 3. NPI glaciers show comparatively less dispersion in the components of the force balance with respect to SPI and, in general, an increase in all components over the period evaluated (Fig. S3, Fig. S4). Our data analysis reveals that marine-terminating glaciers show significant increases in the components of the force balance with respect to lake-terminating glaciers. In this sense, the increase in basal (176.67% versus 17.25%), lateral (255.69% versus 3.87%), longitudinal (256.78% versus 16.14%), and driving
(52.76% versus 42.08%) stresses stand out (Fig. S2). In lake-terminating glaciers, the increase in driving stress over the other stresses is noteworthy (Fig. S2).

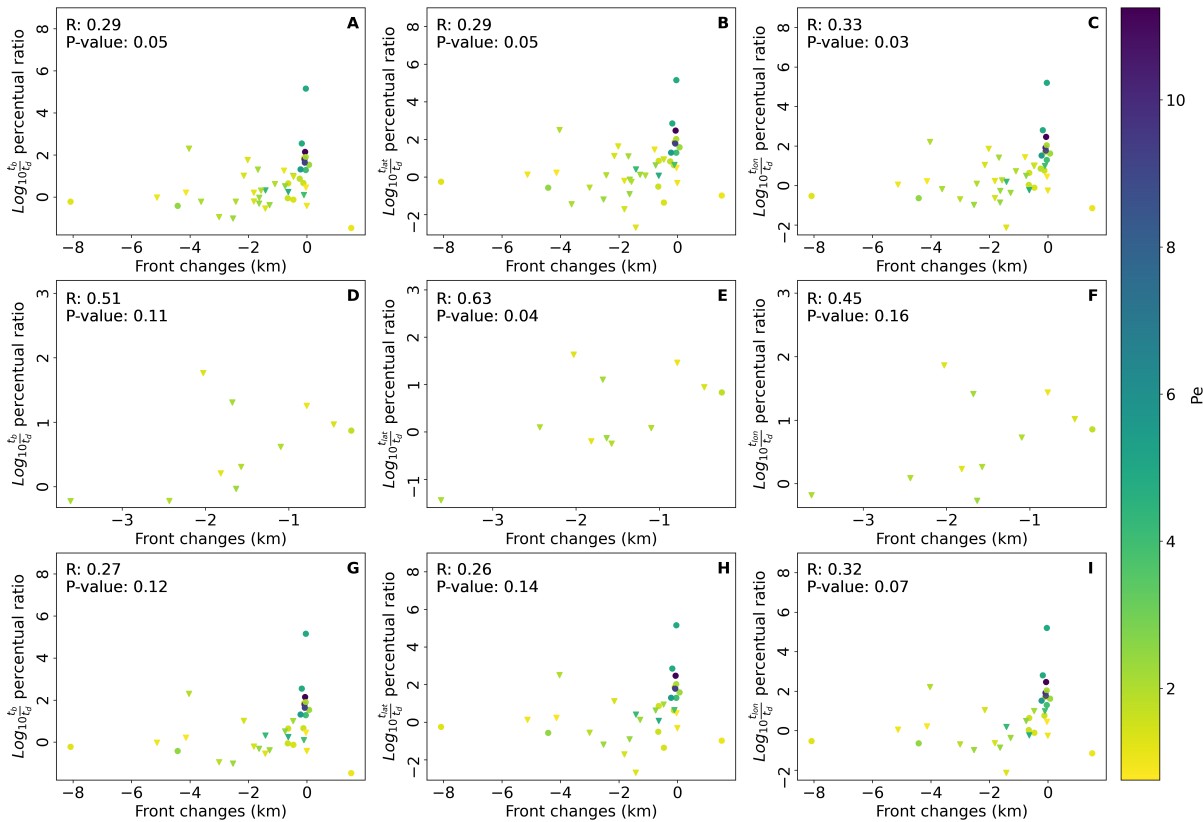

**Figure 7.** Péclet number (median of averages), force balance (median of averages), and terminus changes between 2000-2018 for the first 5 km of the terminus for PI glaciers. Terminus changes in relation to the percentage variation of the log ratio of: A) $t_b$ to $t_d$ ($Log(t_b/t_d)$) for PI glaciers B) $t_{lat}$ to $t_d$ ($Log(t_{lat}/t_d)$) for PI glaciers. C) $t_{lon}$ to $t_d$ ($Log(t_{lon}/t_d)$) for PI glaciers, D) $t_b$ to $t_d$ ($Log(t_b/t_d)$) for NPI glaciers, E) $t_{lat}$ to $t_d$ ($Log(t_{lat}/t_d)$) for NPI glaciers, F) $t_{lon}$ to $t_d$ ($Log(t_{lon}/t_d)$) for NPI glaciers, G) $t_b$ to $t_d$ ($Log(t_b/t_d)$) for SPI glaciers, H) $t_{lat}$ to $t_d$ ($Log(t_{lat}/t_d)$) for SPI glaciers, I) $t_{lon}$ to $t_d$ ($Log(t_{lon}/t_d)$) for SPI glaciers. Marine-terminating glaciers are represented as a circle, and lake-terminating glaciers are described as an inverted triangle.

The results indicate that there are exceptions regarding terminal changes, force balance, and geometric relationships, which we believe are important to highlight. In glaciers that maintain relative stability, we observed that an increase in basal, lateral, and longitudinal stresses significantly compensates for increased driving stress in the terminal region. The ratio of percentage changes of force balance components with respect to driving stress has average values greater than approximately 1.5 in glaciers with average Pe of 4.39 are associated with relative terminus stability (<100 m retreat) (Fig. 7). Therefore, results suggest that when driving stress fails to be supported




by the other components of the force balance glaciers are susceptible to retreat, so the Péclet number is the primary differentiable mechanism in the identified behaviors, giving signs of geometric advantages that favor the evolutionary behaviors of the force balance.

These interpretations are supported by the evolutionary behavior of the Jorge Montt and Penguin glaciers,
marine-terminating glaciers located in SPI, with prograde and retrograde slopes, respectively, and with substantial differences in the Péclet number, whose values are 2.47 and 4.85, respectively. The Jorge Montt glacier has retreated 4422 m versus 178 m for the Penguin glacier. In this case, the Pe limit at Jorge Montt is 18.5 km inland and 6.5 km from Penguin in 2000. Driving stress on the Jorge Montt glacier increased by 435% compared to 2% on the Penguin glacier (Fig. S5). Both glaciers show a substantial increase in basal stress, but more than three
480    times higher in Penguin Glacier, which reaches 831% (Fig. S6). Lateral and longitudinal stress increased in percentage terms in both cases but was comparatively much higher on Penguin Glacier (> 1000%) (Fig. S7, Fig. S8). The magnitude of the increase in stress appears to support the Penguin Glacier's relative stability (or the terminus's transition to such conditions). Thus, Pe influences the ability of Penguin Glacier to evolve into such stress states. However, it also raises the idea that the ability to diffuse thinning depends on the stress state, the
evolution of the stress state, and the ability of the stress state to be transferred upstream (given by geometry or Pe).

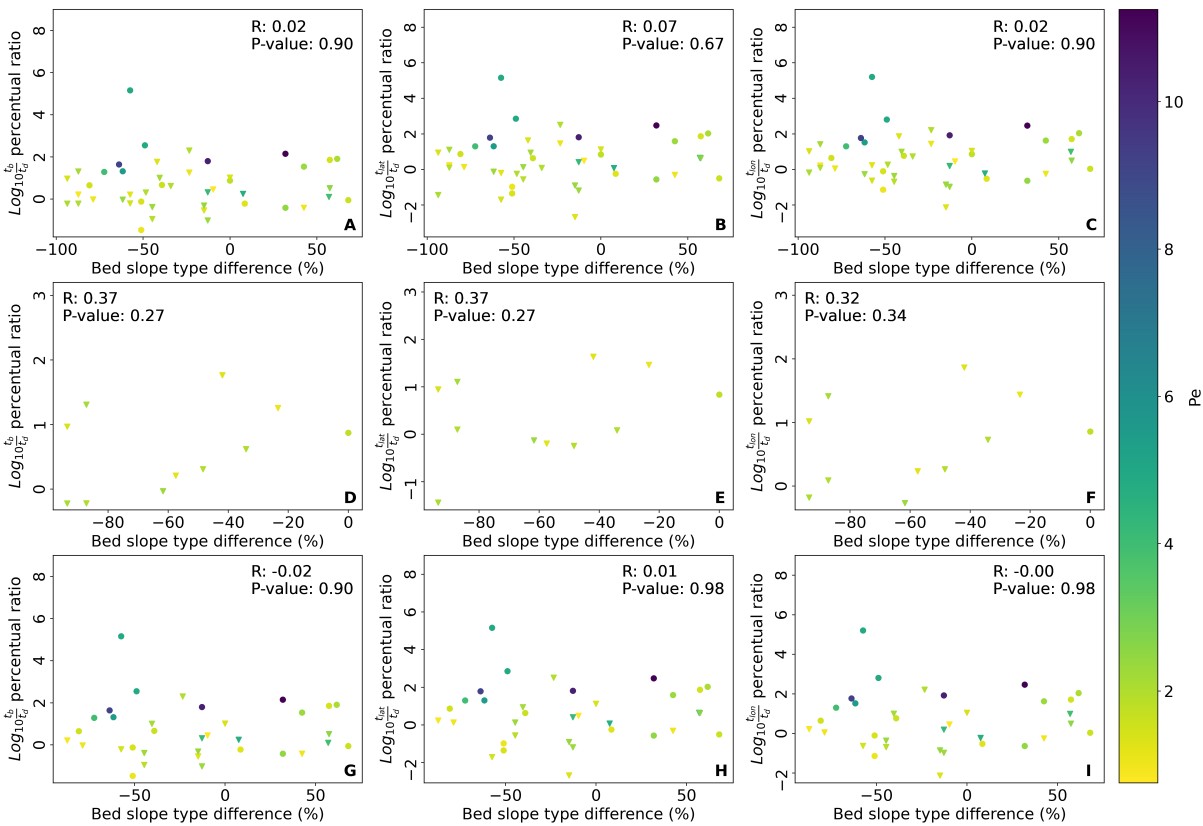

**Figure 8.** Péclet number (median of averages), force balance changes (median of averages), and bed slope type (average) between 2000-2018 for the first 5 km of the terminus. A) Bed slope type in relation to the percentage variation of the log $t_b$ to $t_d$ ratio ($\text{Log}(t_b/t_d)$) for PI glaciers. B) Bed slope type in relation to the percentage variation of the log $t_{lat}$ to $t_d$ ratio ($\text{Log}(t_{lat}/t_d)$) for PI glaciers. C) Bed slope type in relation to the percentage variation of the log $t_{lon}$ to $t_d$ ratio ($\text{Log}(t_{lon}/t_d)$) for PI glaciers. D) Bed slope type in relation to the percentage variation of the log $t_b$ to $t_d$ ratio ($\text{Log}(t_b/t_d)$) for NPI glaciers. E) Bed slope type in relation to the percentage variation of the log $t_{lat}$ to $t_d$ ratio ($\text{Log}(t_{lat}/t_d)$) for NPI glaciers. F) Bed slope type in relation to the percentage variation of the log $t_{lon}$ to $t_d$ ratio ($\text{Log}(t_{lon}/t_d)$) for NPI glaciers. G) Bed slope type in relation to the percentage variation of the log $t_b$ to $t_d$ ratio ($\text{Log}(t_b/t_d)$) for SPI glaciers. H) Bed slope type in relation to the percentage variation of the log $t_{lat}$ to $t_d$ ratio ($\text{Log}(t_{lat}/t_d)$) for SPI glaciers. I) Bed slope type in relation to the percentage variation of the log $t_{lon}$ to $t_d$ ratio ($\text{Log}(t_{lon}/t_d)$) for SPI glaciers. Marine-terminating glaciers are represented as a circle, and lake-terminating glaciers are described as an inverted triangle.





The behavior of the force balance, the Péclet number, and the variations of the terminus differ according to the type of slope (Fig. 8, Fig. S9). Lake-terminating glaciers are characterized by much steeper backward slopes than marine-terminating glaciers, reaching a total of 33.54% and 13.16%, respectively (Fig. S2). On the other hand, low Pe values relate to a predominance of retrograde slopes in the first 5 km of the terminus. Although this condition seems generalized in PI, the Pio XI glacier, characterized by a retrograde slope and low Pe, shows the most significant advances in PI. This suggests an additional mechanism that, despite the predominance of retrograde slopes, can limit the disintegration of the terminus in some cases. On the other hand, the prograded slopes that seem to concentrate less accumulated retreat do not show a sufficiently clear sign of being a sufficient mechanism to mitigate the retreat of the terminus. For example, the prograded slope HPS 12 glacier has the highest recorded retreat in PI. However, in both bed slope configurations, a Péclet number greater than five appears sufficient to reduce term disintegration, along with percentage changes of the force balance components concerning driving stress greater than 1.5.

## 4 Discussion

### 4.1 Empirical limit based on kinematic wave theory and its relation to observed changes

The thinning perturbation propagates diffusively upstream in a large part of the PI glaciers. However, our results indicate that this propagation is limited to values of the Péclet number less than 8. These results align with what has been observed in Greenland glaciers, where an empirical thinning limit was established with Pe < 3 (Felikson et al., 2017, 2021). The differences observed with the limit found in Greenland are likely due to different glaciological and climatic conditions between the two regions, resulting in a higher limit value in PI. Our results highlight the importance of the geometric state as an effective mechanism for detecting areas of vulnerability and, consequently, a metric that can potentially complement studies addressing the impact of anthropogenic climate change projections on glacial evolution. In this sense, Manquehual-Cheuque & Somos-Valenzuela (2021), used a statistical approach to determine regions in Andean Patagonia where glaciers are likely to resist climate change. This study was carried out considering representative concentration paths 4.5 and 8.5. The authors suggested a correlation between glacier persistence and elevation. According to our results, elevation, together with the geometric state, as a consequence of glacier-topography interaction, provides a more complete picture to identify regions where glaciers could resist climate change.





The thinning limit occurs immediately after the bed roughness/bed slope gradient rises and falls due to a knickpoint on the subglacial bed. According to our results, these gradient changes limit the propagation of the thinning from the terminus. Previous research has shown that bed roughness is important in controlling glacier dynamics (Falcini et al., 2022) and ice flow resistance (Hoffman et al., 2022). Roughness is generally found to be associated with the presence of hills, valleys, and bumps under the ice. In this sense, it has been found that bed roughness can contribute to ice flow stabilization (Frank et al., 2022; Diez et al., 2018).

Moreover, Felikson et al. (2021) used a similar approach and found that the thinning limit in Greenland glaciers is associated with large jumps in topography along the ice flow. In the context of PI, Minowa et al. (2023) analyzed the role of topography in the dynamic evolution of lake-terminating glaciers in SPI, stating that before significant mass-loss events, glaciers exhibited super flotation conditions triggering decoupling with topography and driving terminus retreat due to topographic features representing changes in gradient slope/roughness (e.g., over-deepening). In addition to complementing the findings of Minowa et al. (2023), our results suggest that the topography control not only lies in the terminus but also plays an elemental role in the existing inland dynamics.

Our research reveals that on average, about 53% of the ice flow of PI glaciers is located below the vulnerability threshold, due to the position of the empirical thinning limit along the ice flow, indicating a potential risk of future mass loss in these areas. According to our results, the empirical limit is situated at an average distance of 15 km and a maximum of 42 km inland. This is highly complex, considering that these glaciers showed sustained mass loss during the last decades. On the other hand, we found at least two situations (Gualas glacier in NPI and Lucia glacier in SPI) in which the retreat of the term exceeded the position of the empirical limit, warning that the perturbation may extend even beyond the established empirical limit.

## 4.2 Evolution Controlled by Upstream Geometry

We study the geometric influence on the evolution of glaciers in PI between 2000-2018, focusing on the first 5 km (terminus). We found that more than 93% of these glaciers have a Pe $\leq$ 8 in the terminus (year 2000), indicating conditions that favor diffusion of thinning. Glaciers with Pe < 4.85 experience the greatest thinning and retreat; these low Pe values are associated with deeper upstream limits. For example, glaciers with Pe < 4.85 have 59% of their flow below the empirical limit, whereas glaciers with Pe > 4.85 significantly reduce vulnerable flow, averaging approximately 22%. The major exception to the observed pattern in Péclet number is the Pío XI glacier, where we find propagation of thickening of the terminal region in regions with Péclet number $\leq$ 8. Thus, under





certain initial conditions, the Péclet number may indicate a geometric advantage of glaciers to gain mass resulting from the propagation of thickening. However, this may also be due to sudden changes in the glacier basin that are not predictable by the Péclet number. For example, in recent decades, the Pío XI glacier has shown important changes in sediment dynamics that have driven the formation of a large frontal moraine; this condition prevents

direct contact of the glacier with the ocean and, consequently, fluctuations in the ocean-glacier energy transfer (Rivera, 2018). Studies on outlet glaciers recently highlighted by Catania et al. (2020) suggest that the presence of a terminal moraine helps stabilize the terminus because it provides additional resistance to flow (Brinkerhoff et al., 2017; Morlighem et al., 2016), limits access to warm water at depth (Bartholomaus et al., 2013) and reduces the effects of buoyancy during calving (Enderlin et al., 2018; Post et al., 2011). In this regard, Catania et al. (2020)

emphasize that the presence of a moraine at the glacier terminus exerts  a profound control on dynamics that the coupling between ice and sediment dynamics is considered to be solely responsible for the glacier tidal cycle, a pattern of slow advance and rapid retreat of marine glaciers that occurs in the absence of climatic forcing (Brinkerhoff et al., 2017; Post et al., 2011). The disappearance of the conditions that allow the continuous formation of the sediment bar induces the moraine's breakup and prevents the glacier's contact with the ocean,

and, therefore, the calving process could eventually rapidly change the stability conditions. Under these scenarios, the Pío XI could not undergo rapid thinning due to rapid lubrication and the possibility of propagating terminal glacier perturbation upstream (low Péclet number in the first 5 km). The advance-retreat cycle of outlet glaciers has been proposed by research as one of the explanatory mechanisms for the dynamics observed in Pío XI (Rivera et al., 2012; Rivera, 2018) and also in other outlet glaciers in PI that recorded advances and retreats during the

Little Ice Age (Rivera et al., 2012).

The retrograde slope in the dominant condition in glaciers with large retreats in PI, in this sense, our results show that the lowest Pe values are highly connected with the predominance of retrograde slopes (> 50%) in the first 5 km of the terminus, which demonstrates that the vulnerability of these glaciers is also related to the bed geometry. Current large glacier retreats have occurred in regions with a predominance of retrograde slopes,

revealing the importance of this topographic condition in the vulnerability to diffusive thinning. In this regard, research on Greenland outlet glaciers has shown that the geometry of an outlet glacier exerts a first-order control on dynamics by influencing the balance of forces governing ice flow (Van Der Veen and Whillans, 1989; Catania et al., 2020). In multiple cases, the terminus of outlet glaciers resting on retrograde slopes of terminal moraines (termed sills in the oceanographic literature) are generally more susceptible to retreat due to external forcing and,

once retreat begins, may continue unabated until the terminus reaches a region of prograded bed slope (Catania et





al., 2018a; Schoof et al., 2017; Haseloff and Sergienko, 2018). Thirty-year observations of glacier dynamics in Greenland and their interaction with topography using available bed and bathymetry data (Catania et al., 2018) indicated that glacier retreat is accelerated across broad and deep parts of the bed characterized by retrograde bed slopes. Catania et al. (2018) highlight that bed slope and fjord topography are critical controls on terminus

dynamics. Morlighem et al. (2016) sought to quantify the sensitivity and vulnerability of sea-terminating glaciers to ocean-induced melting by studying the dynamics of the Store Gletscher glacier front in West Greenland, and found that ice-ocean interactions are the triggering mechanism for glacier retreat, but the bed controls its magnitude. This is because the glacier breaks away from the sill (sill or terminal moraine) when ocean-induced melting quadruples, at which point the glacier retreats irreversibly for 27 km into a bed of retrograde

characteristics. According to our analysis, lake-terminating glaciers show a predominance of retrograde slopes with respect to marine-terminating glaciers. This condition could drive super-flotation processes and acceleration of ice flow by glacier-topography decoupling in the terminal region, as recently reported for lake-terminating glaciers in SPI (Minowa et al., 2023).

The geometric state of PI glaciers has translated into significant changes in evolutionary terms of the force

balance from 2000 to 2018. In glaciers that maintain relative stability, we observe that an increase in basal, lateral, and longitudinal stress significantly compensates for increased driving stress in the terminus region and is associated with glaciers with Pe > 4.85. Stability conditions in components of the force balance have previously been correlated in areas with a thinning limit close to the terminus region (Carnahan et al., 2022). Our results suggest that glaciers retreat irreversibly when driving stress cannot be supported by the other components of the

force balance. One of the best examples is the HPS 12 glacier, with a retreat of more than 8 km in 18 years. In this sense, Zheng (2022) proposes that glaciers with low Pe result from thick ice and a fast flow, indicating reduced basal friction that can favor an acceleration and thinning of the terminus that can be diffusively propagated upstream.

When analyzing the geometry of the PI glaciers and the type of terminus, we observed that glaciers that end

in the sea and lake have fewer Péclet number than the limit. However, lake-terminating glaciers show a higher diffusivity and percentage of ice flow under the limit than marine-terminating glaciers. Several studies in PI have analyzed the state of disintegration and retreat of lake-terminating glaciers (Irarrazaval et al., 2022; Sakakibara et al., 2013; Minowa et al., 2021, 2023), warning of their rapid retreat in recent decades (Minowa et al., 2021, 2023; Sakakibara et al., 2013). Based on our results, we postulate that the high diffusivity conditions of lake-terminating



glaciers deliver signs that, in the future, they may continue to retreat due to their ability to propagate thinning inland and consequently profoundly reconfigure the PI landscape.

## 4.3 Limitations of our analysis and future work

One of the significant limitations in our research is the lack of quantification of the uncertainty associated with the input data for estimating the Péclet number and the force balance. The velocity products present spatial and
temporal information gaps before the year 2000, mainly due to the low density of satellite data in that period, limiting the possibility of performing a more extensive analysis in time and prior to the date used in this research. To minimize the effects of these limitations, we merged the 1999-2000 satellite product, which allowed us to substantially reduce the number of null pixels (see section 2.2). In PI, there are uncertainties related to bed topography, especially in regions with low observed data density, which limits the quality of the inversion of the
thickness products. Recent estimates of subglacial topography for PI may be appropriate in new experimental approaches (Fürst et al., 2024). Topography uncertainty represents a complex bias, as outlet glaciers are sensitive to topographic variations even at small scales (Catania et al., 2018b, 2020; Enderlin et al., 2013) so uncertainties remain in the projections of the future behavior of the marine- and lake-terminating glaciers. Therefore, progress on ice thickness measurements in PI is required to robust their estimates. On the other hand, it is important to
consider that we do not differentiate the surface mass balance from the elevation change product of Dussaillant et al. (2019), which could introduce an additional bias to our thickness estimates.

Further research should incorporate more topographic elements to obtain a more complete picture of the role of topography in the dynamic evolution of glaciers. Frank et al. (2022) show that changes can be highly complex and depend not only on the topographic conditions at the base but also on the geometry of the fjord. For example,
rapid retreat is more likely to occur in broad and deep fjords, whereas slower retreat is expected in narrow and shallow topography. Our study focused on geometric bed conditions, as the numerical model does not consider other topographic features. In future research, it will be necessary to extend the geometric analysis to elements beyond subglacial topography, including lateral geometric conditions, fjord lithology, and the dynamism of topography in the dynamic evolution of glaciers. This is important because the hardness of topography, which
varies with the geologic setting, can significantly influence glacier dynamics, cause changes in topography in response to these dynamics, and consequently feed new trajectories of glacier change (Bernard et al., 2021). In this regard, signals from a deglaciated region of northwestern Scotland (Bradwell et al., 2019), a significant change was observed about 18,500 to 16,000 years ago, related to the collapse of floating ice sectors and rapid retreat of



ice fronts. This change is attributed to a geological transition from soft to hard bedding and a change in bed geometry, which accelerated the disappearance of ice flow in the region. This highlights the importance of combining bed geometric analysis with lithological variability. On the other hand, ice flow can smooth the bed and consequently generate substantial changes in glacier dynamics (Falcini et al., 2022).

         Our study provides a broad perspective on the influence of glacial geometry on its response to climate. Despite our progress, numerous questions remain about the interaction between glacial geometry and climate variability.
Thus, we see potential for future research in this field that could significantly improve our understanding of the relationship between the thinning limit and climatology. We suggest that future studies consider conducting a more comprehensive analysis of precipitation and temperature trends along flowlines on a more significant number of glaciers and at various time scales to link climate variability and Péclet number.

## 5 Conclusions

We used the Péclet number (Pe), based on the diffusive kinematic wave model, to study the impact of ice flow geometry on the propagation of diffusive thinning and the consequent vulnerability of the 45 major marine- and lake-terminating glaciers in the Patagonian Icefields. According to our data analysis, locations with Pe ≤ 8 were those that suffered the greatest thinning and retreat, suggesting the existence of an empirical limit. This empirical limit encompasses more than 90% of the ice thinning, and is related to a significant change in the slope gradient
and roughness of the subglacial topography in PI due to a knickpoint in the subglacial bed. These findings demonstrate a significant influence of bed morphological characteristics on the thinning limit. They also suggest that on average ~53% of the ice flow of PI glaciers is below the thinning limit.

         Despite the exceptional change in the terminus of the Pio XI Glacier, where the basal conditions changed from a marine-terminating glacier to a land-terminating glacier, these do not ensure its stability in the future. Geometric
conditions and its retrograde bed favor the propagation of diffusive thinning upstream in the event of a change in basal conditions. The lake-terminating glaciers are the most vulnerable to retreat according to their configuration and position of the current front. In NPI, the Pared Norte, Benito and Steffen glaciers stand out. In SPI, the Bernardo, Pingo and Chico glaciers stand out. In addition, the Grosse, Gualas and Colonia glaciers stand out in NPI due to the low number of Péclet in the terminal region. In SPI, the Perito Moreno, Upsala + Cono and
Occidental glaciers. The high diffusivity conditions of lake-terminating glaciers provide signals that, in the future,



they may retreat profoundly due to their ability to propagate their thinning inland, and consequently profoundly reconfigure the PI landscape.

This research demonstrates that geometric state is a key indicator for identifying glaciers vulnerable to thinning. Therefore, it is an essential metric to identify those glaciers that need to be investigated as a priority,
considering current climate change projections. Finally, given the global-scale importance of PI glaciers and their complex vulnerability status, we suggest intensive monitoring of those glaciers showing a high percentage of ice flow below the empirical thinning limit, along with those with low Pe. These glaciers are susceptible to further mass loss in the future.

*Code availability.* The codes generated during the research will be available in Zenodo once peer reviewers review and accept the article.

*Data availability.* The SRTM DEM used to represent the surface elevation is publicly accessible at https://earthexplorer.usgs.gov (last access: 15 March 2024), download platform of the U.S. Geological Survey
(USGS). Data of ice thickness, subglacial topography, elevation and front changes are accessible via QFUEGO-PATAGONIA at https://qfuego-patagonia.org/ (last access: 04 April 2024). Glacier velocity for the years 1999-2000 is accessible from the ITS_LIVE (Inter-Mission Time Series of Land Ice Velocity and Elevation) project website at https://nsidc.org/apps/itslive/ (last access: 04 April 2024; Gardner et al., 2019). Glacier velocity for the year 2018 is accessible at https://doi.org/10.6096/1007 (Millan et al., 2022).


*Author contributions.* BM developed the conceptualization, data collection, curation, data processing, data analysis, and writing – original draft. MSV developed the conceptualization and provided support with supervision, review, editing, and funding. ML support with supervision, review, and funding. II support with review. DF supports with data collection and review. EL supports the review. DR supports with review and
funding. AF support with review and funding.



*Competing interests.* The authors declare no competing competing interests.

*Acknowledgements.* We appreciate the support of the National Geographic research grant EC-95830R-22
(Changing Landscapes: From glaciers to lakes). The first author appreciates the financial support of ANID through
the Beca Magister Nacional 2023 (22231474) and the Master's program in Agricultural Engineering of the
Universidad de Concepción.

*Financial support.* We are funded by the Chilean Science Council (ANID) through the Anillo (ACT210080),
ANID Fondecyt Regular (1230433) and Water Research Center for Agriculture and Mining, CRHIAM
(ANID/FONDAP/1523A0001). The first author was funded by ANID through the Beca Magister Nacional 2023
(22231474).

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
