# Peer review of "Glacier geometry limits the propagation of thinning in Patagonian Icefields"

_EGUsphere, 2024_

## Author Comment (AC1)

**Responses to Comments (C ) Reviewer 1 (R1)**

**General comment R1:** Dear editors and authors,

Thank you for presenting and handling the work about the (in)stability of glaciers in the Patagonian Icefields (PI) (https://doi.org/10.5194/egusphere-2024-1053). I am very excited to see the analysis of Peclet numbers applied for the first time in a region other than the Arctic and the Greenland Ice Sheet. The workflow presented here builds on previously published models with interesting analysis strategies (see my comments below). The results are not easy to comprehend at first sight (for some reasons below), but they reveal patterns that can help decide future plans for monitoring PI glacier changes.

The work is worth publishing in TC to be shared with the glaciology community, especially for those working with PI glaciers or considering using the Peclet number and its model framework on different glacier regions. There are some technical components, however, that I would like to check with the authors and discuss for potential improvement:

**Response to general comment R1:**

Dear reviewer, we sincerely appreciate your valuable comments on our manuscript and recognition of the importance of applying Péclet number analysis to the Patagonian Ice Fields. In the revised version, we have completely restructured the manuscript to improve its clarity and rigor, focusing on three main findings: the identification of the empirical limit of thinning at $Pe \leq 21$, the relationship between Pe maxima and subglacial topographic features, and the evaluation of the vulnerability of glaciers according to the area under the empirical limit. We have strengthened the analysis using the most recent subglacial topography data set from Fürst et al. (2024) and through a robust statistical analysis to validate the empirical limit, in addition to simplifying the presentation of results to highlight the most significant contributions. We believe that these changes have substantially improved the quality and accessibility of the manuscript, maintaining its original scientific contribution to the understanding of the vulnerability of the Patagonian Ice Fields.

**Major comments**

**R1C1:** I might have missed this, but what is the value of l (length of perturbation) as in equation 1?

**Response to R1C1:** In the revised version of the manuscript, we have clarified that l represents the distance from the glacier terminus to any position along the centerline. We have added this clarification explicitly in the methodology section.

**R1C2:** The idea of cumulative thinning (equations 11-12) is okay, but wouldn't it be more physically meaningful if we could link or rename this quantity to the cumulative volume loss since the terminus?

**Response to R1C2:** We agree with your comment. In the revised version of the manuscript, we have substantially modified our methodology to directly relate cumulative thinning to ice volume loss. This is reflected in the new equations 3 and 4, which now explicitly quantify the percentage of volume loss accumulated from the terminus. This is also shown in Figure 2 of the results section.

**R1C3:** I don't quite understand equations 13-14. The x in CT(x) is the distance from the terminus, but in equation 13, you seem to put a Peclet number in replace of x. In addition, CT(x) itself should be a percentage number by definition, but equation 14 sets it equal to Pe_{limit}, which is not a percentage, to my understanding.

**Response to R1C3:** Thank you for identifying our mathematical inconsistencies. In the manuscript's revised version, we eliminated equations 13-14 and simplified our analysis of the empirical thinning limit. We now use only equations 3 and 4, which clearly describe the cumulative thinning from the term as a percentage. Additionally, the identification of the empirical limit ($Pe \leq 21$) is based on a robust statistical analysis demonstrating a significant change in the thinning behavior before and after the identified limit.

**R1C4:** The most necessary improvement in this work is probably the **percentage of ice flow** (equation 15). Unlike the percentage thinning, equation 15 is not physically meaningful to me because ice flow does not add up this way. I was confused multiple times when I read the manuscript; for example, in L552: "glaciers with Pe < 4.85 have 59% of their flow below the empirical limit." (How do you have 59%, not 100%, of the ice flux in a glacier's main trunk? Is the glacier diverging to many branches?) Can you justify the use of this quantity? Alternatively, I can see this quantity might relate to the mapping of high-speed zones of a glacier, and it can physically make sense this way. However, we need a better statement in the manuscript so readers know what we are physically comparing Pe to.

**Response to R1C4:** We sincerely appreciate this critical observation about the incorrect use of the "ice flow percentage" concept. In the revised version of the manuscript, we have corrected this fundamental conceptual error by replacing "ice flow" with "glacier surface area" to correctly refer to the glacier surface area under the empirical thinning limit. For example, we now indicate that "~76% of the glacier surface area is below the thinning limit", which is a physically consistent measurement. This correction has been applied consistently throughout the manuscript, eliminating equation 15 and any reference to percent ice flow that could confuse. The new metric accurately represents the proportion of the glacier area potentially vulnerable to diffusive thinning.

**R1C5:** Figure 2: If this is cumulative thinning (volume loss), why does Pe = 2 have a lower value than Pe <= 1?

**Response to R1C5:** This is due to the variability of the data itself, which does not allow immediate stabilization of the median. This is also observed in Felikson et al. (2017) in the Pe 5-6 transition of Figure 3a. For a more consistent analysis, this new version of the manuscript does address the robustness of the established thinning limit. In this sense, given the model, new data used, and the associated metrics, it is suggested that Pe = 21 is a stable value as a limit.

**R1C6:** For all figures with Pe as the x-axis, we should probably change the tick labels from 1 and 10 to <=1 and >=10, respectively.

**Response to R1C6:** The previously reported values have changed slightly due to the implementation of the new methodological framework. We have adopted a window width of Pe = 1. We also did not include the lower and upper-end values that are not critical for this analysis; for example, values that are smaller than Pe = 0 were not included.

**R1C7:** Since you put p-value in many figures -- Is the "R" correlation coefficient or the coefficient of determination? Is it "R" or "R^2"? What model does the p-value test? It looks like it's testing a linear model, but this should be explicitly specified. And if it's testing a linear model, for trends that are not linear (e.g., Figure 4c and Figure 6b), R(^2) and p-value are not good indicators.

**Response to R1C7:** We appreciate your observation and apologize for the lack of clarity in the statistical analysis. In the manuscript's revised version, we removed linear correlations and p-values that were inappropriate for nonlinear relationships. In the new version of the manuscript, we have implemented a statistical approach focused on identifying the empirical limit of weight loss ($Pe \leq 21$) based on non-parametric metrics such as the median absolute deviation (MAD) and the interquartile range (IQR).

**R1C8:** Some information appears multiple times, such as using QGIS and some Python libraries. Since this is not a short article, I wonder if there's a way to present this once and for all and save the length.

**Response to R1C8:**

Dear reviewer, we appreciate your observation. To avoid redundancy in the description of the tools used, in the revised version of the manuscript, we have consolidated all the information on tools in a single subsection titled **"2.3 Data processing and software"**, where we added the following information to the text:

We processed all data in Python 3.11, using NumPy for numerical operations, Pandas for managing data, and SciPy for statistical analysis and noise filtering. For spatial analyses, we used QGIS 3.22 to manually delineate the glacier centerline, setting a consistent 50-meter spacing along each centerline for data extraction. We also used a Savitzky-Golay filter along the flow direction to reduce noise in thickness, surface elevation, subglacial elevation, and elevation change profiles, following the methodology of Zheng (2022).

**R1C9:** L600-601: The force balance model already assumes that tau_d is the sum of the three resistive forces (cf. L244-245). So what do you mean by "compensate" here?

**Response to R1C9:** Thank you for observing the inconsistent terminology in our analysis of the balance of forces. In the previous version of the manuscript, when we talked about "compensating" in percentage terms, we were referring to the relative changes in the components of the force balance over time. This means that if the driving stress ($\tau d$) increases by a certain percentage, the resistive stresses ($\tau b$, $\tau lat$, $\tau lon$) must be adjusted collectively in a proportion that maintains the equilibrium of the system. However, we recognize that the term "compensate" was not the most appropriate term in this context.

In the revised version of the manuscript, we have entirely removed text related to force balance to maintain focus on our main finding on the empirical thinning limit based on the Péclet number. We believe this decision has allowed us to present a more coherent and direct analysis of the relationship between glacial geometry and the spread of thinning.

**R1C10:** The authors plan to release the code after the paper is accepted. Do you also plan to release the derived data set, such as the Peclet numbers along the selected flowlines? I appreciate it if you could specify this in the Data Availability section.

**Response to R1C10:** We have released the code associated with our analysis and the data set used, including the central lines analyzed in this manuscript version. We hope that with this, anybody can produce our results quickly and in a friendly manner, in addition to facilitating the application of the methodology in unexplored regions.

**Minor comments**

**R1C11:** There are also a few copyediting suggestions as below:

**Response to R1C11:** We have carefully checked your recommendations for the text. However, due to the length of the article and the request for systematization by the reviewers, some of the suggestions could not be included since the associated texts were eliminated. We specify whether this situation applies in the text for each of your comments.

**R1C12:** L129: annual precipitation?

**Response to R1C12:** Indeed, there are PI areas that exceed 10,000 mm per year. Given the length of the study area description, the paragraph in question was excluded from the subsection.

**R1C13:** L289: The percentage of total thinning is expressed as the thinning at position $x$ along the flowline as follows

**Response to R1C13:** In the manuscript's revised version, we have reformulated this section, presenting equations 3 and 4 more clearly and precisely. We now explicitly explain that the cumulative thinning is calculated as a percentage of the total thinning along the centerline, where x represents the distance from the terminus. Additionally, the mathematical notation has been simplified to make the physical interpretation of these equations more accessible.

**R1C14:** L435: the vulnerability of these

**Response to R1C14:** Dear reviewer, thank you for bringing this error to our attention. The text in question was removed from the current version of the manuscript due to the request for length reduction.

**R1C15:** L600: glaciers that are/have been relatively stable

**Response to R1C15:** Because the force balance was removed from our analysis, the text in question was eliminated in the new version of the manuscript.

**R1C16:** L610: I don't understand what it means to "have fewer Péclet number than the limit." Less Péclet number? What limit do you mean?

**Response to R1C16:** Dear reviewer, thank you for pointing out this ambiguity. In the original text, we referred to the fact that glaciers that end in the ocean and lakes presented, on average, in the first 5 kilometers from their front, an average value of Pe lower than the empirical limit we had defined at that time as Pe = 8. However, in the current version of the manuscript, we have completely restructured the results and discussion, eliminating this specific analysis of the terminal region to focus on the new empirical limit identified (Pe ≤ 21) and its relationship with the observed changes in the glaciers, which provides a more precise and more direct presentation of our main findings.

**R1C17:** L634: The geometry of the fjord is also a topographic condition?

**Response to R1C17:** In technical terms, the shape of the fjord is an intrinsic topographic condition in a glacial environment, which influences the dynamics through multiple mechanisms (e.g., Frank et al., 2022). In our analysis, the possible impact of the morphological characteristics of the fjord is excluded; instead, we rely exclusively on the subglacial topography.

References:

Frank, T., Åkesson, H., De Fleurian, B., Morlighem, M., & Nisancioglu, K. H. (2022). Geometric controls of tidewater glacier dynamics. *The Cryosphere*, *16*(2), 581-601.

**R1C18:** L663: The statement here has nothing to do with basal conditions. Maybe "Despite Pio XI Glacier changed its terminus state from marine to land..."?

**Response to R1C18:** There was indeed confusion in our writing since we were referring to the change in status of the front of the Pio XI glacier. However, in the current version of the manuscript, and as part of the overall restructuring to improve clarity and focus, this sentence has been removed from the conclusions, which now focus on our main findings on the empirical limit of thinning and its implications.

**R1C19:** L815: Two Felikson et al. (2021) references.

**Response to R1C19:** We have corrected the duplicate reference.

---

## Author Comment (AC2)

**Responses to Comments (C) Reviewer 2 (R2)**

**General comment R2:** This paper investigates the influence of glacier geometry on the recent thinning and retreat of ocean/lake terminating outlet glaciers in the Patagonia Icefields. To analyze the geometrical control on glacier changes, the authors compute the Peclet number along the flowline of 45 outlet glaciers. This approach has been proposed by a previous study (Felikson et al., 2017) and applied to glaciers in Greenland and in Svalbard (Felikson et al., 2017; 2020; Zheng, 2022).

Ocean and lake terminating glaciers in Patagonia are rapidly losing mass under the influence of increasingly negative mass balance as well as the ice-ocean/lake interaction. This is a similar situation as in Greenland, although the climate and glaciological settings in these regions are substantially different. Therefore, the application of the recently proposed analysis to Patagonia is interesting and potentially important to better understand the current and future mass loss of the glaciers. As far as I know, this is the first time that Peclet numbers are analyzed for glaciers in Patagonia.

Despite the novelty and potential importance of the study, it is difficult to understand the findings and implications of the study. The way of presentation is one reason, but I suspect fundamental problems in some of the analyses. The manuscript suffers from unclear text and equations, which get in the way of understanding. I list below my major concerns, which are followed by specific/minor comments on the manuscript.

**Response to general comment R2:** We thank the reviewer for valuable comments and for recognizing the novelty and potential importance of our study on the influence of glacial geometry on the recent thinning and retreat of marine- and lake-terminated glaciers in the Patagonian Ice Fields.

In the revised version of the manuscript, we have made substantial changes to address concerns raised about the clarity and rationale of the analysis. We completely restructured the methodology and results sections to provide a more precise and straightforward presentation, focusing on our main finding: identifying the empirical thinning limit at Pe $\leq$ 21, which encompasses more than 95% of ice thinning. In addition, we significantly strengthened the database by using the more recent and robust set of subglacial topography developed by Fürst et al. (2024), which incorporates more than 1.4 million thickness observations.

The analysis has been simplified and strengthened by using single centerlines instead of averages of multiple streamlines, incorporating statistical analyses to validate the robustness of the empirical boundary, and establishing more explicit connections between glacier geometry and observed thinning patterns. Figures have been redrawn to illustrate our main findings directly, and the discussion has been refined to focus on the most significant implications of our results.

These changes have resulted in a more precise and focused manuscript, which we believe effectively communicates our main findings and their implications for understanding the current and future evolution of the Patagonian Ice Fields.

**Major comments/concerns**

**R2C1:** Data analysis and presentation. The authors analyze and present the general trends and statistics obtained from 45 outlet glaciers. Except for maps showing some numbers for each glacier (Figures 5 and 6), readers are not able to see data obtained for each individual glacier. Considering the diversity of glaciers in Patagonia as well as large uncertainty in the bed elevation, showing only statistical values is not convincing and insufficient to draw conclusions. I encourage the authors to look into the details of each individual glacier as performed in previous studies (Felikson et al., 2017; 2020; Zheng, 2022). The Peclet number is a value computed from glacier geometry and ice dynamics. To use it as a measure of glacier stability, investigation of the observational data used for the computation (bed and surface elevation, ice speed, elevation change) along the flowline is necessary (e.g. Figure 2 in Felikson et al., 2017).

**Response to R2C1:** We appreciate your feedback, which has allowed us to reflect the results obtained for each glacier analyzed more appropriately. Due to this, we have substantially modified our analysis and presentation of results. In the revised version, we provide a detailed analysis of individual glaciers. We also changed the data sources used, for example, using the more recent and robust data set of Fürst et al. (2024). We also used the model for Pe developed by Felikson et al., 2017 and expanded in Felikson et al., 2021, which does not use surface velocity data. The latter was done because the new versions of the updated velocity products, mainly during 1999-2000, were more strict in quality control, which generated a significant increase in null pixels in regions where we previously had data. This led us to use an approximation that does not directly include this information since these new data made an appropriate analysis impossible. This dataset represents a significant advance in the accuracy of subglacial topography for the Patagonian Ice Fields, reducing previously noted uncertainties.

Following the methodology of Felikson et al. (2017), we have included the analysis of profiles along the central lines showing Pe, surface elevation, basal elevation, and elevation changes. The above has been incorporated into new appendices, the content of which can be easily reproduced using the codes developed in our research, which are available in Zenodo.

**R2C2:** Bed elevation data. In comparison with Greenland and Svalbard, observations of glacier bed elevation are sparse in Patagonia, and thus, subglacial geometry has a more significant uncertainty. Therefore, a more careful analysis is required for the bed estimated by inversion (Farinotti et al., 2019). As suggested above, investigation of the glacier cross-section and observations along the flowline of each glacier is necessary. As indicated by the authors (Line 625), please also consider using a more recently compiled bed elevation data set (Furst et al., 2024).

**Response to R2C2:** We appreciate this critical observation regarding the elevation data of subglacial topography. In the current version of the manuscript, we have addressed this concern using the most recent dataset developed by Fürst et al. (2024), which represents a significant advance in the characterization of the subglacial topography of PI. This dataset incorporates over 1.4 million thickness observations to produce the most advanced basal topography map to date. The quality of these data is substantially superior to previous products because they include local measurements that have not been published. Our entire analysis has been recalculated using this new dataset, which provides a more solid basis for our study of the relationship between glacier geometry and thinning propagation. The above has been

incorporated into new appendices, including longitudinal profiles, the content of which can be easily reproduced using the codes developed in our research, which are available on Zenodo.

**R2C3:** Force budget. I understand that the components of the force budget (Equations 3-5) were computed from surface strain rates obtained from ice speed maps. The authors assume full slip condition referring to a previous study (Line 261, Collao-Barrios et al., 2018), but the previous study suggested 98% sliding specifically for the fastest-flowing glacier tongue of San Rafael Glacier. It is not realistic to assume 100% slip condition for all the glaciers and regions extending upglacier.

Further, the presentation and discussion of the force balance analysis are difficult to understand. The authors argue that "glaciers retreat irreversibly when driving stress cannot be supported by the other components of the force balance" (Line 604, similar statement in Line 470). What do you mean by "cannot be supported"? The driving stress should be supported by other stress components as stated in Line 244-245. Actually, "the basal drag is calculated as the residual" (Line 267), thus I assume imbalance does not happen.

Changes in the force balance components near the glacier fronts between 2000 and 2018 are presented in page 23-24. Some numbers show very large changes, represented by >1000% increase in lateral and longitudinal stresses at Penguin Glacier (Line 481). Without detailed analysis, such a rapid change is difficult to accept and cannot be simply connected to the argument "the increase in stress appears to support the Penguin Glacier's relative stability" (Line 482).

**Response to R2C3:**

We appreciate your important comments on the force balance analysis in our manuscript. After carefully considering these and the other reviewers' comments, we have decided to remove the force balance analysis from the study for several fundamental reasons.

First, we recognize that the assumption of complete sliding based on the specific observations of the San Rafael Glacier is inappropriate for the entire set of glaciers studied. The study by Collao-Barrios et al. (2018) refers specifically to the tongue of the San Rafael Glacier.

In this sense, our original methodology for the force balance adopted an SSA (Shallow Shelf Approximation) type approach with basal drag without including internal deformation. Through this, we sought to capture the dynamics of temperate outlet glaciers, where basal drag longitudinal and lateral stresses can be significant; however, extrapolating a condition of complete basal drag inland on all glaciers might not be appropriate.

Due to the above, we have excluded the force balance from our analysis and focused the manuscript on our main finding: identifying and characterizing the empirical thinning limit based on the Péclet number. This decision has allowed a more straightforward presentation of our main results. As part of future work, we suggest exploring the relationship between the force balance and the observed Pe values using a more rigorous methodological framework that can adequately capture the complex dynamics of the Patagonian Ice Fields glaciers.

**R2C4:** Percent ice flow. It is hard to understand the "the percent ice flow" defined by Equation 15. CT is "the cumulative thinning from the glacier front in percent" (Line 295), thus Equation 15 gives a mean of CT between the front and the empirical thinning limit. Why do you call this value the percent ice flow? I am not able to follow the analysis and discussion of this value.

**Response to R2C4:**

We thank the reviewer for highlighting this important conceptual ambiguity in our original manuscript. In the revised version, we substantially modified the terminology and the analysis to avoid this confusion.

We have eliminated the "percent ice flow" concept and replaced it with a more precise and physically meaningful metric: "percent glacier surface area." This change is not merely terminological but more accurately reflects what we are quantifying: the proportion of glacier surface area below the empirical thinning limit (Pe $\leq$ 21).

The equations and analysis have been updated to reflect this conceptual clarification in the revised manuscript.

**R2C5:** Discussion of the data and results. It is a pity that the conclusion of the paper tells not much more than "90% of the ice thinning is occurring below the locations with Pe<8" (Line 657-669). This is because the complex data sets presented in the "3 Results" section are not properly discussed in the "4 Discussion" section. In "4.1 Empirical limit ...", no clear interpretation is given to the relatively large Peclet number found for the upper limit of the thinning. "4.2 Evolution Controlled by Upstream Geometry" discusses the advancing Pio XI Glacier, but difficult to find the point of the argument from previous studies in other regions (Line 549-575). The rest of the section describes previous studies and the discussion is not based on the results obtained in this study. Except for the section "4.3 Limitations of our analysis and future work", the discussion is not well-performed and fails to draw conclusions.

**Response to R2C5:**

Dear reviewer, we thank you for pointing out these shortcomings in the discussion and conclusions of our original manuscript. In the revised version, we have entirely restructured these sections to provide a deeper and more meaningful interpretation of our results. The discussion section was completely restructured to provide a deeper understanding of our main results and their implications for PI evolution. We have also developed a more robust discussion of the topographic control on thinning propagation. We have ensured that each section of the discussion is firmly anchored in our results, clearly interpreting their meanings and contextualizing them within the existing literature rather than simply describing previous studies. This restructuring has resulted in a more cohesive and meaningful discussion. The conclusions have been expanded to highlight our main findings. In particular, we emphasize how the current geometric configuration of glaciers makes them particularly susceptible to climate change, with significant implications on water resources, ecosystems, sea level rise, and landscape evolution.

**Specific/minor comments:**

**R2C6:** Line 21: "Gt year-1 annually" is redundant.

**Response to R2C6:** We have corrected this expression in the text, removing the word "annually."

**R2C7:** Line 33: complex "changes in" stresses...?

**Response to R2C7:** We have modified the text; the sentence now reads, "When a glacier retreats, the ice at the terminus experiences complex changes in stresses, including longitudinal, transverse, and shear stresses."

**R2C8:** Line 45: "melting caused by water body-glacier contact" >> "melting due to upwelling plume"?

**Response to R2C8:** We appreciate your comment. We were indeed referring to "melting due to upwelling plume." We have corrected the text.

**R2C9:** Line 65: The statement "3 mm increase..." is after Zemp et al. (2019).

**Response to R2C9:** We added the following reference to the text.

Zemp, M., Huss, M., Thibert, E., Eckert, N., McNabb, R., Huber, J., Barandun, M., Machguth, H., Nussbaumer, S.U., Gärtner-Roer, I., et al., 2019. Global glacier mass changes and their contributions to sea-level rise from 1961 to 2016. Nature568, 382–386. https://doi .org /10 .1038 /s41586 -019 -1071 -0.

**R2C10:** Line 82: "terrain elevation values" >> Do you mean "surface elevation"?

**Response to R2C10:** The term was corrected by surface elevation.

**R2C11:** Line 84: What do you mean by "characteristic terrain elevation values"? Definition?

**Response to R2C11:** The text refers to the dominant elevations in the region; for clarity, we have replaced the text in question with "Surface elevations predominantly range," which more clearly reflects what we wanted to express.

**R2C12:** Line 95-96: Please consider significant digits of these numbers in %.

**Response to R2C12:** Changes were made to the text according to the recommendations.

**R2C13:** Line 101: "quickest" >> fastest flowing?

**Response to R2C13:** We appreciate your comment and have changed the text as recommended.

**R2C14:** Line 102: "second largest" >> San Quintin is the largest in the Northern Patagonia Icefield. Do you mean "second fastest"?

**Response to R2C14:** We have changed the text as recommended.

**R2C15:** Line 106-121: I wonder if reviewing such details and numbers is necessary for this paper.

**Response to R2C15:** Indeed, extending the description of the study area too much does not significantly impact our research. We have removed the associated lines from the study area subsection.

**R2C16:** Line 135-146: Please also consider more recent studies, e.g., Sauter et al., 2020 (Hydrol. Earth Syst. Sci.), Wiedemann et al., 2018 (Front. Earth Sci.) and Salazar et al., 2024 (Climate Dynamics).

**Response to R2C16:** We appreciate your recommendation. However, given the length of the study area description, the paragraph in question was excluded from the subsection. This was done considering your comments and those of reviewer three.

**R2C17:** Line 190: "q is the ice flow" >> ice flux?

**Response to R2C17:** We have corrected the associated text.

**R2C18:** Line 190: the "surface" slope of?

**Response to R2C18:** We have corrected the associated text.

**R2C19:** Line 199: "upper current" >> upglacier?

**Response to R2C19:** We have corrected the associated text.

**R2C20:** Line 225-226: "We used the NumPy, Pandas, and SciPy libraries..." >> This or similar statements are repeated many times.

**Response to R2C20:** We have modified the related text. In order to specify this information only once, we added the following subsection to the text:

"2.3 Data processing and software

We processed all data in Python 3.11, using NumPy for numerical operations, Pandas for managing data, and SciPy for statistical analysis and noise filtering. For spatial analyses, we used QGIS 3.22 to manually delineate the glaciers' centerline, setting a consistent 50-meter

spacing along each centerline for data extraction. We also used a Savitzky-Golay filter along the flow direction to reduce noise in thickness, surface elevation, subglacial elevation, and elevation change profiles, following the methodology of Zheng (2022)."

**R2C21:** Line 236: "... between 2000 and 2018 in PI glaciers ..." >> Not only here, but "PI glaciers" is not necessary.

**Response to R2C21:** During the restructuring of the manuscript, we sought to reduce the use of 'PI glaciers.'

**R2C22:** Equations 11 and 12: The notations are odd. Do you mean something like this? $dH(x_i) = -dh(x_i)/\{\sum_{j=1}^{N} -dh(x_j)\}$

**Response to R2C22:**

We thank the reviewer for pointing out the confusing notation in equations 11 and 12. In the revised version of the manuscript, we have simplified and clarified these equations following his suggestion. The new equations in section 2.5 (equations 3 and 4) now clearly express how we calculate the cumulative thinning from the terminus to any position x along the centerline.

**R2C23:** Equation 14: The Left-hand side is cumulative thinning, whereas the right-hand side is the Peclet number. I am confused.

**Response to R2C23:** Thank you for pointing out the inconsistency of equation 14. Indeed, the original equation was conceptually confusing, complicating the interpretation of our methods and results. In the revised version of the manuscript, we have completely rewritten the associated methodology section, removed the related equation, and reformulated it completely.

**R2C24:** Line 317: "slope gradient" >> Do you mean "slope"?

**Response to R2C24:** This referred to the 'bed slope gradient.' This was specified below: For that, we quantified multiple topographic parameters: bed elevation, bed slope (first derivative of elevation), and bed slope gradient (second derivative). The characterization also included analyzing the distribution of prograde (positive) and retrograde (negative) slopes with respect to the empirical thinning limit.

**R2C25:** Line 323-324: What do you mean by "slope difference"?

**Response to R2C25:** The proportion of prograde and retrograde slopes along the centerline, in this case, the first 5 km upstream from the glacier terminus.

**R2C26:** Line 331: "we classified ..." >> This is already described before (Line 274 and 298).

**Response to R2C26:** We appreciate your comment; now, we only specify this information once in section "2.5 Empirical thinning limit", as shown below: we group the cumulative percentage thinning values in windows of width Pe = 1.

**R2C27:** Line 342-343: The first two sentences are not necessary.

**Response to R2C27:** The mentioned text has been removed due to the new structure of the results section.

**R2C28:** Figure 2: Please use the same scale for the vertical axes of the two plots. I wonder why accumulated thinning (%) decreases upglacier from Pe=1 to 2 (Figures 2a and b) and Pe=8 to 9 (Figure 2a).

**Response to R2C28:** This is due to the variability of the data itself, which does not allow for the immediate stabilization of the median. This is also observed in Felikson et al. (2017) in transition 5-6 of Figure 3a. Our new analysis addresses the robustness of the established thinning limit. Given the metrics, Pe = 21 is a stable value as a limit.

**R2C29:** Line 343: The first sentence in the figure caption is not necessary. Should be in the main text.

**Response to R2C29:** Due to the new structure of the results section, the mentioned text has been removed.

**R2C30:** Line 371: What do you mean by "force balance increases"? It's not a quantity.

**Response to R2C30:** We thank you for pointing out this conceptual inaccuracy. The phrase "force balance increases" is indeed incorrect since force balance is not a quantity that can increase or decrease but rather a state of equilibrium between different stress components. In the associated text, we referred to the percentage variations of the different stresses analyzed along the flow lines. It is important to note that in the context of restructuring the manuscript, we have eliminated the analysis of force balance as a consequence of what was discussed in R2C3.

**R2C31:** Line 386: "thinning limit" >> Is this the same as "empirical limit"? Please be consistent.

**Response to R2C31:** The terms refer to the empirical limit of thinning. Although it continues to be used interchangeably as a writing strategy to avoid excessive repetition of the term, we seek to ensure that this occurs where it does not confuse.

**R2C32:** Line 394: What do you mean by "geometric limit"?

**Response to R2C32:** The term referred to the 'empirical limit of thinning.' The phrase was rewritten in the context of restructuring the results section.

**R2C33:** Line 397-405: Please consider the significant digits for the numbers.

**Response to R2C33:** Dear Reviewer, We have corrected all associated data by including only the appropriate significant digits.

**R2C34:** Figure 5E: I am confused by the different scales given to the three maps.

**Response to R2C34:** Dear reviewer, incorporating different scales could have been confusing. In this new version of the manuscript, we did not incorporate additional maps to the study area (Figure 1), as we believe it is unnecessary to present our results properly.

**R2C35:** Line 418: "Fig. 8" >> Which plot in Figures 8A-I supports this statement?

**Response to R2C35:** We thank the reviewer for pointing out this lack of precision in the reference to the figures. In the original manuscript, the reference to Fig. 8 did not specify which specific panel supported our claim about Pe values. In that context, panels D-F and G-I allow the differentiation for NPI and SPI to be visualized, respectively. The color bar represents the variability of Pe. However, as part of the overall restructuring of the manuscript, we have significantly simplified the presentation of our results and the associated figures. In particular, we have removed Figure 8. The figures in the revised version of the manuscript have been redesigned to more directly and clearly illustrate our main findings.

**R2C36:** Line 427: "Fig. 8" >> Isn't it Figure 6? I am confused.

**Response to R2C36:** Dear reviewer, indeed, the plotted Pe in both cases was the same, i.e., the median for the first five kilometers from the upstream front. However, in Fig. 8, the force balance was additionally plotted. In the context of text restructuring, the figures in the revised version of the manuscript have been removed.

**R2C37:** Line 529: "According to our results," >> Which results? This is one example that I find difficulty in following the discussion. Please support your argument with data.

**Response to R2C37:** We thank the reviewer for pointing out this lack of clarity in referencing our results. In this case, we referred to Figure 3 and the associated text described in the results section. We sought to connect our discussion to our findings in the current manuscript version.

**R2C38:** Line 658: "suggesting the existence of an empirical limit" >> Pe $\leq$ 8 was obtained by setting 90% as the threshold of so called empirical thinning limit. Why Pe $\leq$ 8 suggests the existence of such a limit?

**Response to R2C38:** In our previous analysis, we set Pe = 8 because more than 90% of the cumulative thinning from the glacier front occurred below this value. In the current version of the manuscript, we have significantly strengthened the identification of the empirical limit (Pe $\leq$ 21) through a statistical stability analysis. This analysis shows that the variability of the data after Pe = 21 is substantially reduced, with the median absolute deviation and interquartile range decreasing to approximately one-third of their pre-limit values. The confidence intervals

also narrow considerably after this threshold, showing that thinning statistically stabilizes for observations of Pe greater than the established limit.

**R2C39:** Line 673-674: I understand the potential importance of the analysis, but difficult to follow this conclusion. What is "geometric state"? Which data show it is "a key indicator" and "essential"?

**Response to R2C39:** Dear reviewer, in the revised version of the manuscript, we have significantly strengthened this final section by providing a more precise definition and specific evidence supporting our conclusions. The "geometric state," characterized by the Péclet number, represents the spatial configuration of the glacier resulting from its interaction with topography. Our results demonstrate its importance as a key indicator, where $Pe \leq 21$ values encompass more than 95% of ice thinning, with statistical indicators that strengthen these findings. The practical relevance of this indicator is illustrated with concrete examples from our analysis. For example, we show that, on average, 76% of the surface area of glaciers in the Patagonian Ice Fields is below this empirical limit, with glaciers highly susceptible to thinning propagation and consequent mass loss. This quantification of vulnerability to thinning provides signals of glaciers that might require priority monitoring in the current context of climate change.

**R2C40:** Line 788-793: What is the difference between 2018a and 2018b?

**Response to R2C40:** We appreciate you pointing out this inconsistency in the bibliographical references and have corrected the text.

**R2C41:** Line 812-817: Duplicated?

**Response to R2C41:** We appreciate you pointing out this inconsistency in the bibliographical references and have corrected the text.

---

## Author Comment (AC3)

**Responses to Comments (C) Reviewer 3 (R3)**

**General comment R3:** The manuscript by Morales and others represents a major and impressive amount of work, with a thorough assessment of the interactions between glacier geometry and past and future change of the Patagonian Icefields. To my knowledge, the work is unique and novel, and, through coupling Peclet number analysis with force balance analysis, is innovative in its approach. The authors link the rapid decline of some glaciers within the Patagonian Icefields to their geometric properties.

However, despite the significance and extent of this work, several factors limit the impact of this work and prevent me from endorsing it for publication in its current form.

**Response to general comment R3:**

Dear reviewer, we sincerely appreciate your comments on our manuscript and the recognition of the novelty and importance of our research on the interaction between glacial geometry and changes in the Patagonian Ice Fields (PI).

To address the identified limitations that affect the work's impact, we have substantially restructured the manuscript and updated our experiments based on newly available data. First, we have significantly simplified the presentation of results to focus on three main findings: (1) the identification of the empirical thinning limit at Pe $\leq$ 21, (2) the relationship between Pe maxima and subglacial topographic features, and (3) the assessment of glacier vulnerability based on the area under the empirical limit.

The methodological basis of the study has been strengthened by incorporating the most recent subglacial topography dataset from Fürst et al. (2024), which includes over 1.4 million thickness observations. Furthermore, due to the recent update of existing velocity products for PI and the consequent loss of valid velocity pixels for the year 2000, we decided to implement the Pe solution developed by Felikson et al. (2021) that does not use observed velocity data. We simplified our analysis using centerlines instead of multi-streamline averages and incorporated statistical analysis to validate the identified empirical boundary.

A complete verification and restructuring of the methodology and results sections has improved the presentation of the manuscript. Figures have been redesigned to illustrate our main findings directly, and the discussion has been refined to establish more explicit connections between glacier geometry and observed thinning patterns. This is also reflected in the new supplementary material.

We believe that these changes have substantially improved the clarity and rigor of the manuscript while maintaining its original scientific contribution to understanding how glacier geometry controls thinning propagation in the Patagonian Ice Fields. Finally, we make the data and associated codes available for the full reproduction of our analysis.

References:

Felikson, D., A Catania, G., Bartholomaus, T. C., Morlighem, M., & Noël, B. P. (2021). Steep glacier bed knickpoints mitigate inland thinning in Greenland. *Geophysical Research Letters*, *48*(2), e2020GL090112.

Fürst, J. J., Farías-Barahona, D., Blindow, N., Casassa, G., Gacitúa, G., Koppes, M., ... & Schaefer, M. (2024). The foundations of the Patagonian icefields. *Communications Earth & Environment*, *5*(1), 142.

**Major comments/general points**

**R3C1:** The methods section is hard to follow, so it is difficult for me to know what exactly is being presented in the results section. I don't believe I would be able to reproduce this work. For example, I'm not sure how the authors assign a single Pe to an entire glacier (or region of a glacier?), as in Figure 6E.

**Response to R3C1:** We thank the reviewer for this critical comment on methodological clarity. We have completely restructured the methodology section to improve its reproducibility and clarity. Significant changes include a complete reorganization of section 2, now divided into logical subsections that present the data sources (2.2), data processing (2.3), calculation of the Péclet number (2.4), the empirical thinning limit (2.5), and the relationship between topography and glacier geometry (2.6).

In particular, we have clarified the process of calculating the Péclet number, specifying that it was performed along the centerlines of each glacier at 50-meter intervals. Data extraction and processing are now explained step-by-step, including the use of the Savitzky-Golay filter to reduce noise in the profiles.

The analysis of the empirical thinning limit has been mathematically detailed by equations 3 and 4, which describe how cumulative thinning and its percentage are calculated. We have also clearly explained how we identified the local maxima of Pe.

In a specific subchapter, we include all the tools and libraries used: Python 3.11, NumPy, Pandas, SciPy, and QGIS 3.22. In addition, we release all the codes generated to reproduce our results and analysis. This makes the study transparent and guarantees complete reproducibility, facilitating future research that seeks to apply and/or expand our analysis.

**R3C2:** The results presented are so extensive and thorough that it is hard to identify the specific patterns that the authors intend to draw the readers' attentions to. The same is true for the Study Area section. Within the scatter plots of Figs 5-8, there are so many plots, and most of them do not appear to be discussed at all. I recommend that the authors identify just those critical pieces of information necessary to support their discussion and conclusions sections and eliminate the remainder. Superfluous information, while potentially valuable for some purposes, distracts from the main points and "muddies the water," reducing the impact of your paper.

**Response to R3C2:** We appreciate this important observation regarding the presentation of results. We have substantially simplified the presentation of our findings, removing extraneous information and focusing on the critical patterns that support our main conclusions. In particular, we have significantly reduced the number of figures and graphs, keeping only those

that directly illustrate our identification of the empirical thinning limit (Pe $\leq 21$) and its relationship to subglacial topography.

The results section now more clearly and directly presents key patterns: the distribution of cumulative thinning relative to Péclet number, the relationship between Pe maxima and bed characteristics, and the vulnerability of glaciers according to the area under the empirical limit. Each remaining figure is fully discussed in the text and contributes directly to our main conclusions, thus improving the clarity and impact of our findings.

Additionally, we have condensed the study area section to include only the contextual information essential to understanding our results, removing secondary details that do not directly contribute to our research objectives.

**R3C3:** At the same time, some of the results discussed in the text, like the percentage of the modern Patagonian Icefields that are susceptible to future diffusive thinning, do not appear supported by any figures. I'm not sure where to find this valuable information.

**Response to R3C3:** Dear reviewer, the relevant information was represented in Figure 5 of the old version of the manuscript. However, we agree that this information was not appropriately represented. Due to the new structure used in the results section, these data are specified in Figure 3. In this new figure, we explicitly added the area under the empirical limit for each of the glaciers with a limit found. Additionally, in the released codes, it is possible to directly reproduce the values mentioned in the current text of the manuscript.

**R3C4:** Given the length of the manuscript and the need to better highlight/focus on particularly critical scientific results, it may well be necessary to cut significant portions of the text, and perhaps even analyses, to keep the length reasonable and not distract the reader with non-essential detail.

**Response to R3C4:** We appreciate this important suggestion regarding the length and focus of the manuscript. We have substantially restructured the results section to emphasize the most critical scientific findings. In particular, we have reorganized the section to focus on two key results: (1) the identification of the empirical thinning limit at Pe $\leq 21$, which encompasses more than 95% of ice thinning, and (2) the strong correlation between Pe maxima and subglacial topographic features.

We removed analyses related to the balance of forces and focused our analysis on thinning behavior and frontal changes with respect to the Péclet number. We believe this restructuring allows for a straightforward presentation of our scientific contributions and a more efficient address of our research objectives.

**R3C5:** The fluency of the writing is not where it needs to be. I suggested grammatical edits for the first page or so before I stopped. The entire manuscript needs careful editing for clarity of language.

**Response to R3C5:**

In response to your feedback, we have conducted a thorough review of the manuscript to improve the clarity and fluency of the writing significantly. The implemented changes include a complete restructuring of sections to improve the logical flow of the text, particularly in the

methodology and results, simplifying complex sentences, and removing redundancies to enhance comprehension. In addition, we conducted a detailed review of grammar, punctuation, and sentence structure.

**R3C6:** Throughout the manuscript, "ice flow" is used where some form of ice amount seems implied, like glacier surface area. Please correct.

**Response to R3C6:** We thank the reviewer for this important observation. We have carefully reviewed the entire manuscript and replaced all instances where 'ice flow' was incorrectly used to refer to glacier area or extent. Specifically, we now use 'glacier surface area' when referring to the spatial extent of glaciers below the thinning limit. For example, we modified the text to read '...on average ~76% of the glacier surface area of PI glaciers is below the thinning limit' instead of using 'ice flow.' This change has been applied consistently throughout the manuscript to improve clarity and technical accuracy.

**R3C7:** In light of these limitations, I do not feel like I was able to properly evaluate the present work. I could not follow the chain from methods to results to discussion. As such, I would not consider this a complete and thorough review. After revision, the manuscript will require a fresh and complete review.

**Response to R3C7:**

We appreciate your honest assessment of the limitations of conducting a full manuscript review. We understand that the lack of clarity in the connection between methods, results, and discussion significantly hampered the evaluation of our work. In response to this concern, we have substantially restructured the manuscript to establish a clear and logical sequence between these critical sections. The changes include a complete reorganization of the methodology, now divided into well-defined subsections, a significant simplification of the results section to focus on the most relevant findings, and a more focused discussion that connects directly to these results.

**R3C8:** I wish I could be more encouraging as to the state of the present work, and I applaud the ambition of the current study. My best wishes to the authors.

**Response to R3C8:**

Dear reviewer, we sincerely appreciate your recognition of the ambition of our study and your good wishes. We believe that your constructive and detailed comments, together with those of the other reviewers while identifying important areas for improvement, have been instrumental in significantly increasing the quality of our manuscript.

**Specific/minor comments/Line edits:**

**R3C9:** L 10: Some ambiguity-- gradient of slope of the bed, or of the surface? If surface, I think this would be more clear if this was the "gradient of surface topography." Unless you intend to actually refer to the topographic curvature, as in the gradient of the gradient of surface elevation (which is what you've written)?

**Response to R3C9:** In this case, we were referring specifically to the bed slope gradient, corresponding to the second derivative of bed elevation. We have modified this information throughout the text to avoid confusion.

**R3C10:** L 11-12: Why 53% of "ice flow," Do you just mean surface area? How do you quantify a percent of ice flow? What's the baseline for that ratio?

**Response to R3C10:** We thank the reviewer for pointing out this ambiguity. We were indeed referring to glacier surface area and not ice flow. We have corrected this throughout the manuscript, including the abstract, where we now state that '~76% of the glacier surface area' lies below the empirical thinning limit. This percentage represents the proportion of the total glacier area that lies between the terminus and the empirical thinning limit (Pe ≤ 21). For clarity, we have rewritten these sections to specify that we are referring to glacier surface area, indicating the distances inland where this limit lies (on average 25 km and a maximum of 59 km from the terminus). The calculation is based on the ratio of the cumulative area from the terminus to the empirical limit divided by the total glacier area, thus providing a straightforward quantification of what proportion of each glacier is potentially vulnerable to diffusive thinning.

**R3C11:** L 17-18: "allows us to project" makes it seem like you're taking credit for this work.

**Response to R3C11:** We thank the reviewer for pointing out this inaccuracy in the wording, and we have corrected the text.

**R3C12:** L 20: "that glaciers there have lost"

**Response to R3C12:** Hemos corregido la frase asociada.

**R3C13:** L 26: specify what types of "contributions" you're referring to.

**Response to R3C13:** We modified the text to explicitly specify that we are referring to their contribution to sea level rise on a global scale, rewriting the sentence as "their contribution to sea level rise on a global scale."

**R3C14:** L 27-28: Glacier geometry plays a much broader role on ice flow than is supported just by these manuscripts. Broader lit review is necessary. Even to a textbook.

**Response to R3C14:**

Dear reviewer, we have added the following basic references:

- Nye, J. F. The mechanics of glacier flow. *Journal of Glaciology*, *2*(12), 82-93. 1952.
- Paterson, W. S. B. *Physics of glaciers*. Butterworth-Heinemann. 1994.
- Hooke, R. L. *Principles of glacier mechanics*. Cambridge University Press. 2005.

**R3C15:** L 29: access of meltwater to what?

**Response to R3C15:** We have modified the text to specify that we are referring to meltwater access to the glacier bed. The sentence now reads, "This geometric control influences the access of meltwater to the glacier bed and consequent basal lubrication," which explicitly clarifies that

meltwater accesses the glacier bed, which is critical to understanding its role in basal lubrication.

**R3C16:** L 30: the "balance of forces" is nearly synonymous with ice flow dynamics. Why is this point different than the citations for ice flow dynamics?

**Response to R3C16:** We thank the reviewer for highlighting this conceptual redundancy. In the revised version, we have removed the reference to the "balance of forces" and reorganized this section to focus specifically on the distribution of stresses. The references cited in this section (Catania et al., 2020; Enderlin et al., 2018; Pfeffer, 2007; Van Der Veen, 1996) focus specifically on studies that analyze how glacier geometry affects the spatial distribution of these stresses and their consequences on glacier stability, thus differentiating themselves from previous references that deal with general ice flow dynamics.

**R3C17:** L 32: Much better would be to cite some of the old Nye or Harrison papers that are first cited in Felikson 2017, for example "The response of glaciers and ice-sheets to seasonal and climatic changes" by Nye in 1960.

**Re**sponse to R3C17: Nye (1960) was cited and referenced in the main text.

Reference:
Nye, J. F.: The response of glaciers and ice-sheets to seasonal and climatic changes, Proc R Soc Lond A Math Phys Sci, 256, 559–584, https://doi.org/10.1098/rspa.1960.0127, 1960.

**R3C18:** L 37: "glaciers in West Greenland" or "glaciers in western Greenland"

**Response to R3C18:** Text corrected.

**R3C19:** L 38: "it has been found" è use active voice

**Response to R3C19:** The change was made using active voice.

**R3C20:** L 39: revise typos

**Response to R3C20:** We have verified and corrected the typos.

**R3C21:** L 40: "overdeepened regions"

**Response to R3C21:** The grammatical correction was made in the sentence.

**R3C22:** L 41: I think you mean retreat "extent" (as in length), not duration as in time.

**Response to R3C22:** We appreciate your correction; we did indeed refer to 'extent' in terms of length. We have corrected the text.

**R3C23:** Here, I'm ceasing line edits. There are too many to track and suggest edits for each one. This manuscript will need thorough and complete editing for grammar and clarity.

**Response to R3C23:** Dear Reviewer, We have thoroughly reviewed the text for grammar and clarity to provide the most refined version possible.

**R3C24:** L 63: In referencing potential for sea level rise, cite the original papers that the IPCC cites, not the IPCC itself.

**Response to R3C24:** Dear reviewer, we have added the following references: Dussaillant et al., 2019; Braun et al., 2019; Millan et al., 2019.

Referencias:

Dussaillant, I., Berthier, E., Brun, F., Masiokas, M., Hugonnet, R., Favier, V., Rabatel, A., Pitte, P., and Ruiz, L.: Two decades of glacier mass loss along the Andes, Nat Geosci, 12, 802–808, https://doi.org/10.1038/s41561-019-0432-5, 2019.

Braun, M. H., Malz, P., Sommer, C., Farías-Barahona, D., Sauter, T., Casassa, G., Soruco, A., Skvarca, P., and Seehaus, T. C.: Constraining glacier elevation and mass changes in South America, https://doi.org/10.1038/s41558-018-0375-7, 1 February 2019.

Millan, R., Rignot, E., Rivera, A., Martineau, V., Mouginot, J., Zamora, R., Uribe, J., Lenzano, G., De Fleurian, B., Li, X., Gim, Y., and Kirchner, D.: Ice Thickness and Bed Elevation of the Northern and Southern Patagonian IcefieldsPI, Geophys Res Lett, 46, 6626–6635, https://doi.org/10.1029/2019GL082485, 2019.

**R3C25:** L 67: here again is ice flow, when I think you mean ice field, as in "area" (not motion).

**Response to R3C25:** Dear reviewer, indeed, as you mentioned and we have clarified in the previous comments, we are referring to 'glacier surface area.' We have corrected the text to read "that glacier geometry limits the propagation," removing the incorrect reference to "ice flow."

**R3C26:** L 120: This is starting to feel like a laundry list of various glacier facts, without a framework for identifying what are the most important points. Can you synthesize more so as to better call attention to what facts are most relevant for the paper?

**Response to R3C26:** Dear reviewer, indeed, an extremely extensive description of the study area was not appropriate, which does not generate any significant positive impact on our research. Due to the above, we have reduced the study area subsection. We appreciate your comment that allowed a more concise description of the study area.

**R3C27:** L122-146: Again, this is a lot of information, and I, the reader, don't yet have any real context as to what of this is necessary or how it would slot into the paper. This section should all be motivated by the Intro. I don't see how it is.

**Response to R3C27:** We thank the reviewer for highlighting this important structural issue. We agreed that the section contained excess information without a clear connection to the study's objectives. In response to his comment, we have completely restructured the section by removing non-essential details.

**R3C28:** Figure 1: This is a beautiful map. Nice going.

**Response to R3C28:** Dear reviewer, thank you for your feedback.

**R3C29:** L 164: Why not use the glacier thickness data from Millan 2022? In that paper, they show that it performs much better than the Farinotti dataset for icefields, and so is much more appropriate here.

**Response to R3C29:** Dear reviewer, we re-did all our calculations using the recent model released by Furst et al. (2024), which was calibrated with local measurements that had not been published until then; now, our results are based on the state-of-the-art model for the region.

Referencias:

Fürst, J. J., Farías-Barahona, D., Blindow, N., Casassa, G., Gacitúa, G., Koppes, M., ... & Schaefer, M. (2024). The foundations of the Patagonian icefields. *Communications Earth & Environment*, *5*(1), 142.

**R3C30:** L 180: I'm not clear on what it means to use the gradient of the velocity as a mask for flowlines. Also, I would call these "profiles" rather than flowlines if they're manually delineated and not actually following flow. Why aren't they just following flow? That would be cleaner and preferable to using manually delineated profiles. I've written a tool to extract flowlines from any velocity dataset, and it's here: https://github.com/tbartholomaus/project-tools

Maybe it's useful to you?

**Response to R3C30:** Dear Reviewer, we appreciate this valuable comment. Indeed, the generation of the flowlines could have been done by developing an appropriate algorithm such as the one you provided. However, we believe that due to the discharge in different directions of some of the glaciers (commented by Reviewer 1) and because of what is raised in R3C43, using the flowlines could make our analysis more complex. Considering the above, we believe that adopting the central line is a more appropriate approach. This approach has been previously implemented in large-scale analyses (for example, Felikson et al., 2017). Additionally, we have replaced all references to 'flowlines' with 'central line' in the text.

References:

Felikson, D., Bartholomaus, T. C., Catania, G. A., Korsgaard, N. J., Kjær, K. H., Morlighem, M., Noël, B., Van Den Broeke, M., Stearns, L. A., Shroyer, E. L., Sutherland, D. A., and Nash, J. D.: Inland thinning on the Greenland ice sheet controlled by outlet glacier geometry, Nat Geosci, 10, 366–369, https://doi.org/10.1038/ngeo2934, 2017a

**R3C31:** L 190: specify "surface slope"

**Response to R3C31:** We have specified the information in the text.

**R3C32:** L 191: partial with respect to x needs no definition

**Response to R3C32:** We eliminate the definition of the partial derivative with respect to x.

**R3C33:** L 193: "motion of the perturbation" as a whole is more than just advection.

**Response to R3C33:** Dear reviewer, thank you for the clarification. To avoid any confusion, we have removed this information from the text.

**R3C34:** L 197: "length"? do you mean span? Or distance from the terminus?

**Response to R3C34:** Dear reviewer, thank you for pointing out this ambiguity. We have corrected the text to specify that "l" represents the distance from the glacier terminus to any position along the centerline.

**R3C35:** L 199: What is "upper current"?

**Response to R3C35:** We thank the reviewer for pointing out this error. The term "upper current" was a translation error. The term is intended to refer to ice flow in the upglacier regions. In the revised version of the manuscript, we have replaced "upper current" with "upglacier regions."

**R3C36:** L 205-206: Are these characteristic dimensions of some sort, or dimensions at a specific location?

**Response to R3C36:** We thank the reviewer for highlighting this lack of clarity. In the revised version of the manuscript, we have clarified that the variables $\alpha_0$ and $H_0$ represent the surface slope and thickness of the glacier, respectively, measured at each point x along the centerline.

**R3C37:** L 206: is $U\_0$ the average speed? Sliding speed?

**Response to R3C37:** Dear reviewer, thank you for this observation. The highlighted text corresponds to the average velocity of the glacier, given the methodological basis of the approach used. In the revised version of the manuscript, this specification is no longer necessary since we have adopted the model developed by Felikson et al. (2021) to calculate the Péclet number, which does not require surface velocity data. This methodological decision was made due to the limitations found in the updated velocity products for 1999-2000. These present a significant increase in null pixels in regions where we previously had data, making an appropriate and consistent analysis difficult.

**R3C38:** L 213: How did you combine data from the full glacier lengths to arrive at an average Pe value for each glacier?

**Response to R3C38:** This operation involves averaging the mapped flowline indices. For example, the average of the first six vertices of each flowline. We believe that this may lead to spatial discrepancies that could affect our analysis. Therefore, we decided to redo our analysis using only the central line of the glacier.

**R3C39:** L 224: What are "distance units"?

**Response to R3C39:** Dear reviewer, thank you for pointing out this inaccuracy in the specification of distance units. In the revised version of the manuscript, we have replaced the ambiguous phrase "distance units" with "We extracted elevation changes at 50-meter intervals along the centerline".

**R3C40:** L 229: Do you mean Meier et al. as the authors, or are you referring to yourselves?

**Response to R3C40:** We appreciate your comment. In the associated text, we did not refer to "we"; it was related to Meier et al. (2018). We have explicitly modified the text by adding this information: Meier et al. (2018) delineated the glaciers through a semi-automated process using satellite images from the Landsat and Sentinel constellations for this last inventory.

**R3C41:** L 261: I'm not aware of this reference, but surely there's some caveat to this. 98% sliding can't be true throughout the entire icefield.... Even up at high elevation, slow-flowing areas? Even on the Greenland Ice Sheet, at a spot with about 100 m/a flow, internal deformation is 30-50% of the surface speed. I don't think this need throw off your entire analysis, but it seems important to recognize and consider the implications.

**Response to R3C41:** Dear reviewer, we thank you for this important observation regarding the inadequate generalization of basal sliding conditions. We acknowledge that our extrapolation of the 98% basal sliding observed at San Rafael Glacier (according to Collao-Barrios et al., 2018) to the entire ice field was poorly constrained, especially considering the spatial variability of dynamics, climate, and topography.

In the revised version of the manuscript, given the associated comments from Reviewer 2, we have removed the force balance analysis and focused on the study of glacier geometry and its relationship to thinning propagation.

**R3C42:** L 266: the value you give for gravity is an acceleration, not a force.

**Response to R3C42:** We sincerely regret this error and have removed this inaccurate reference as part of a broader restructuring that excludes balance of power analysis from our research.

**R3C43:** L 270: Why is this averaging approach? Also, what if the flow lines branch and either stay apart or re-converges around a nunatak? Then the different lengths of flowlines at a given locale might be offset, and the averaging would be inappropriate.

**Response to R3C43:**

Thank you for pointing out these important limitations related to averaging multiple streamlines. We agree that the original approach of averaging six streamlines could introduce significant biases, especially in cases where streamlines diverge around nunataks or have different lengths, resulting in averages not representative of actual glacier conditions. In response to this concern, we have substantially modified our methodology in the revised version of the manuscript. Specifically, we have corrected our analysis based on the centerline. Please review our response to R3C30.

**R3C44:** L 271: Filtering in the along-flow direction?

**Response to R3C44:** We thank the reviewer for requesting this clarification. In the revised version of the manuscript, we have explicitly specified that the Savitzky-Golay filter is applied along the direction of the glacier centerline, specifically in section 2.3 (Data processing and software).

**R3C45:** L 275: Are these Pe classes just rounded Pe values? If so, why introduce a different term than just the Pe values themselves? If not, how does this differ from rounded Pe values?

**Response to R3C45:** Dear reviewer, thank you for pointing out the inappropriate use of Pe classes. We have substantially simplified our approach in the revised version of the manuscript. Instead of out-of-range value classes, we now use moving windows with a fixed width of Pe = 1. Please see section 2.5 (Empirical thinning limit), where our methodology is clarified.

**R3C46:** L 279-280: No part of this sentence makes sense to me. Please find another way to convey your intention.

**Response to R3C46:** In the revised version of the manuscript, we have removed this confusing sentence entirely. This removal is part of a broader decision to remove the balance of forces analysis from the manuscript.

**R3C47:** L 291: These 'd's are typically differential operators and should not be used otherwise. For changes or differences, I recommend using capital Deltas.

**Response to R3C47:** In the revised version of the manuscript, we have adopted the notation $\Delta$ to represent differential.

**R3C48:** L 292-293: Odd mixing of actual coordinates ("x"s) and indices in this terminology. Obscures meaning.

**Response to R3C48:** We have unified the notation to use exclusively spatial coordinates, where x consistently represents the distance from the glacier terminus measured along the centerline. This simplification eliminates the ambiguity that you have noticed.

**R3C49:** L 293: How is position N decided?

**Response to R3C49:** We have eliminated the index-based notation (including the N position) and instead used a more unambiguous notation based on the distance from the glacier terminus. The landward position is now expressed simply as the distance x from the terminus, measured at 50-meter intervals along the glacier centerline, thus avoiding confusion from using indices. This change simplifies the mathematical notations described in the methodology to make our analysis comprehensible.

**R3C50:** L 296: Again, while your code is probably written with respect to indices, finding indices in your equations is unnecessarily confusing.

**Response to R3C50:** We have simplified the mathematical notation to make it more intuitive and straightforward, eliminating excessive indices. This change simplifies the mathematical notations described in the methodology to make our analysis as comprehensible as possible.

**R3C51:** L 292-301: I'm really not following these methods. Can you revise the text to clarify? Or include a figure? On further consideration, the notation in equations 11 and 12 seems more complicated than necessary. Maybe reviewing with a mathematics colleague might help you find a more straightforward way of writing this out.

**Response to R3C51:** Dear reviewer, In the revised version of the manuscript, we have substantially simplified this section, replacing equations 11 and 12 with a new set of equations (3 and 4) that more directly and clearly describe the calculation of cumulative thinning.

**R3C52:** L 298: How do you define these Pe classes?

**Response to R3C52:** Dear reviewer, the classification or grouping of the data was done using a Pe = 1, except at the extremes, where values less than or greater than 0 and 10 were included, respectively. In the methodology section, we directly specify the mechanisms used for its classification. This new approach does not include values out of range at the extremes.

**R3C53:** L 301: There's a little too much in this sentence to follow. Can you break into smaller pieces to make it easier to digest?

**Response to R3C53:** Dear reviewer, we have rewritten the associated section in order to make it easier to read.

**R3C54:** L 312: I think you mean percent of area, not percent of ice flow.

**Response to R3C54:** Dear reviewer, we were indeed referring to that. We have corrected all the associated text throughout the manuscript.

**R3C55:** L 314: I'm finding this page quite difficult to follow. I don't really understand how you're running your analysis, and what the variables you're producing and discussing later in the paper will be.

**Response to R3C55:** We apologize for the lack of clarity in the presentation of our methodology. In the revised version of the manuscript, we have completely restructured the methodological section to provide a clear and logical sequence of our analysis.

**R3C56:** L 317: By slope gradient, do you mean the second derivative of the surface elevation? As in the terrain curvature, or the change in surface slope? Or do you just mean the surface slope?

**Response to R3C56:** Dear reviewer, thank you for pointing out this ambiguity in our terminology. In the revised version of the manuscript, we have clarified the definition and usage of these slope-related terms. Specifically, we have modified section 2.6 to clearly distinguish between bed slope, which is the first derivative of bed elevation with respect to horizontal distance, and bed slope gradient, which is the second derivative of bed elevation and represents the rate of change of slope, effectively being a measure of bed curvature.

**R3C57:** L 319: How do you turn prograde and retrograde slopes into percents?

**Response to R3C57:** From the front, we calculated the mean subglacial elevation gradient (in the previous analysis, the average of the indices of the six flowlines). Once calculated, regions with negative gradients were classified as retrograde and positive as prograde. The percentage corresponded to the proportion of positive and negative observations along the "mean"

flowline. The conversion to percentages is not performed in this new manuscript version. Moreover, only the central line of the glacier is used.

**R3C58:** L 336: What is an "analysis of terminus"?

**Response to R3C58:** In the old version of the manuscript, reference was made to the specific analysis of the first five kilometers of the "average" flow line from the glacier front. This is no longer done now, according to the new approach adopted, which has been previously specified.

**R3C59:** L 344-351: Unfortunately, I don't quite understand the methods sufficiently to interpret this figure.

**Response to R3C59:** Dear reviewer, we hope that the new structure of the methodology section will provide a clear perspective of the analysis mechanisms that led to these latest results.

**R3C60:** L 367: Please introduce the force balance results generally first, before diving into their changes and how changes in force balance coincide with changes in geometry.

**Response to R3C60:** In the revised version of the manuscript, we have removed the force balance analysis to focus on our main findings related to the control of glacier geometry on thinning propagation.

**R3C61:** L 374-375: How is it possible for all the resistive forces to drop but the driving stress to not drop? Otherwise, the forces are not in balance. The fact that the basal drag is the residual requires that the forces stay balanced and I'm not sure how this statement could make sense. This same point persists in Fig. 4. How can the driving stress increase, but all of the resistive stresses drop?

**Response to R3C61:** Dear reviewer, thank you for your question. In this case, we were referring to percentage variations, not absolute ones. This could have generated the conditions described in the previous version of the manuscript. It is important to note that this is not analyzed in this version of the manuscript due to the synthesis of information that has been carried out, so all the associated text has been eliminated.

**R3C62:** L 376-377: This last sentence more appropriately belongs in the discussion section.

**Response to R3C62:** Dear reviewer, we have followed your recommendations during the restructuring of the manuscript.

**R3C63:** Figure 4: What are the R and p-values here? I wouldn't have thought there should be a good basis for a linear relationship between these two terms. What's the physical justification? If there isn't a physical justification, then take these terms off.

**Response to R3C63:** Dear reviewer, we agree on the absence of a physical justification for a linear correlation analysis and p-value of these variables. Considering the above, we have eliminated all figures and associated analysis, considering the context of restructuring the manuscript.

**R3C64:** L 385-386: Can you include a plot to show these relationships? Again, I suspect you're using "ice flow" when you really mean "ice area"?

**Response to R3C64**: The requested information is now specified in Figure 3. Indeed, as we have previously mentioned, the ice area was correct.

**R3C65:** Figure 5: Are all the symbols individual glaciers? The last line of the caption implies it. Why are there no symbols for land-terminating glaciers?

**Response to R3C65:** Our research did not assess land-terminating glaciers. As mentioned in section 2.1, our analysis focuses on glaciers with areas larger than 65.5 km², representing more than 80% of the total area of the Patagonian Ice Fields. Land-terminating glaciers were not included in the analysis since they represent a marginal fraction of the total area (only 2% in the Southern Ice Field and 18% in the Northern Ice Field, according to the data presented in the study area section) and typically correspond to smaller glaciers that do not meet our area selection criteria.

**R3C66:** L 387: I don't understand this statement. Isn't the empirical thinning limit an up-glacier location, where the Pe value exceeds some threshold (threshold of 8, in this case)? If so, how can the glacier have retreated past it? This isn't the empirical limit for retreat, is it?

**Response to R3C66:** In the experiment presented in the previous manuscript, we believe this may have occurred because 1) we did not base our experiments on maximum Pe and 2) changes in temporal gradients in mass balance affect the stability of the boundary. However, it is essential to note that no glacier has retreated beyond the newly established boundary in this manuscript version, providing signals that it is a robust value.

**R3C67:** L 398: Inappropriate, or unnecessary, 4 significant figures presented. Here and all the other four significant figure numbers in this paragraph.

**Response to R3C67:** Dear Reviewer, We have corrected all associated data by including only the appropriate significant digits.

**R3C68:** L 415-416: What does this sentence mean?

**Response to R3C68:** We thank the reviewer for this observation. Indeed, the highlighted sentence came from an earlier approach to our manuscript, where we analyzed the behavior of the first 5 km of the terminal region in terms of Péclet number, elevation changes, frontal changes, and force balance. This analysis indicated that 93% of the glaciers in the terminal region had a Pe less than eight. However, in the current version of the manuscript, we have substantially modified our approach, focusing on the identification of the empirical thinning limit (Pe $\leq$ 21), its landward position, and the evaluation of the glacier response in terms of frontal changes, thus eliminating the specific analysis of the terminal region and making the presentation of our main results more straightforward.

**R3C69:** L 418: Is Figure 8 mentioned here before Figs 6 and 7? All figures should be numbered according to the order they appear in the text.

**Response to R3C69:** In the revised version of the manuscript, we have completely reorganized the figures to ensure that their numbering corresponds to the order in which they appear in the text. The former Figure 8 has been removed as part of our restructuring to provide a more focused presentation of the main results. We have carefully reviewed the entire manuscript to ensure that figure references are consistent with this new order.

**R3C70:** L 420-421: Are these the lowest Pe values of this icefield? What do you mean as "standing out?" With respect to what?

**Response to R3C70:** They correspond to the lowest magnitude median Pe observed in marine-terminating glaciers in Campos de Hielo Sur, specifically for the first five kilometers from their front. In the revised version of the manuscript, we have eliminated this characterization of the Pe values in the terminal region. This modification is part of our restructuring that attempts to present the results more clearly and directly, focusing on the patterns of glacier vulnerability identified through the analysis of Pe along the entire central line and not in a specific region.

**R3C71:** L 419-426: Maybe it would be useful to plot or just make a table of these numbers? In paragraph form, I'm not sure what of these numbers are particularly important or noteworthy. Perhaps you could put all the numbers in a table, and then just synthesize them a little more to highlight the particularly important results.

**Response to R3C71:** Instead of listing all the values in paragraph form, we present this information in Figure 3, which graphically shows the distribution of vulnerable area per glacier, the distance to the empirical limit, and the differentiation between marine and lake terminus glaciers.

**R3C72:** L 427: Write out "retreat and thinning," and perhaps first identify within the results section that thinning and retreat co-occur.

**Response to R3C72:** In the revised version of the manuscript, we have explicitly indicated the co-occurrence of these phenomena. The results section has been completely rewritten.

**R3C73:** Figure 7: The font sizes for the subscripts of the different stress components are too small to read.

**Response to R3C73:** We acknowledge your comment about the readability issues in Figure 7. However, as part of the manuscript's overall restructuring, we have removed Figure 7 and the associated force balance analysis. In the current manuscript version, we have maintained the appropriate font size in the figures.

**R3C74:** Figure 7: I'm afraid I don't quite understand what is meant by a "median of averages" for the Peclet numbers. But to interpret, I need to know where these Pe values are for- and what they represent for each glacier. I assume this is for some region near the terminus? But I didn't understand the methods sufficiently to get this.

**Response to R3C74:** The expression referred to the median values of the average of the six flow lines in the first five kilometers from the front of each of the glaciers evaluated. In the previous version of the manuscript, we sought to understand in more detail how the geometry of the region near the terminus modulated the propagation of thinning upstream. This analysis is no longer performed in the new version of the manuscript, so the associated information was removed.

**R3C75:** Figure 7: There is so much information in this figure but it is barely discussed in the text. Maybe only at line 470? Cut out any information that is not discussed in the text and essential to your discussion points/conclusions, as set up by your introduction section.

**Response to R3C75:** Considering the previously described changes, this information was removed entirely. Throughout the text of the new version, we have referenced the figures coherently.

**R3C76:** L 470-471: Because glaciers are an example of Stokes flow, the stresses on a parcel of ice must always balance. If you're finding that forces do not balance, either you're violating Stokes flow and the principles of a force balance, or I'm missing an important piece of your analysis.

**Response to R3C76:** In the previous discussion, we referred to the percentage increase of specific components of the force balance; in this context, the balance between them is maintained. In the revised version of the manuscript, we have decided to eliminate the force balance analysis. We decided to focus on the empirical thinning limit based on the Péclet number ($Pe \leq 21$) and the consequent vulnerability for the PI glaciers.

**R3C77:** L 477: Inappropriate number of digits again.

**Response to R3C77:** Dear reviewer, thank you for pointing out this error. In the revised version of the manuscript, we considered significant digits appropriate to the context.

**R3C78:** L 483: Or, I might write that "The high Pe value of Penguin Glacier reflects geometric factors that allow Penguin to adjust its force balance through an increase in basal drag, without retreating."

**Response to R3C78:** We appreciate your suggestion to improve the explanation related to Penguin Glacier. However, in the revised version of the manuscript, we have substantially modified our approach by removing the force balance analysis to focus on the direct relationship between glacier geometry and thinning propagation.

**R3C79:** L 528-533: These are important discussion points, but the figures that demonstrate this are not clear. The important results are buried in too complete a representation of all the different results of your study. Strip down information content throughout the paper so that the important results shine, and set you up for your discussion section, like here.

**Response to R3C79:** We agreed that a more precise presentation of our main results was needed. In the revised version of the manuscript, we have substantially restructured both the results section and the discussion to highlight our most important findings. In this context, we have significantly reduced the number of figures, keeping only those that directly illustrate our main findings: Figure 2, which shows the identification of the empirical limit at $Pe \leq 21$; Figure 3, which illustrates the spatial distribution of vulnerable areas, and Figure 4, which demonstrates the relationship between Pe and subglacial topographic variability.

The discussion section has been reorganized to focus on three key aspects: 1) the interpretation of the empirical limit of $Pe = 21$ and its comparison with previous studies, 2) the relationship between Pe maxima and glacial bed characteristics, 3) and the implications for the future vulnerability of Patagonian glaciers. We removed secondary analyses that detracted from our main messages.

**R3C80:** L 541: Again, this is a result that I would really like to see conveyed in a convincing, elegant, clear figure. If you've shown it already, I'm afraid that I've missed its demonstration.

**Response to R3C80:** In response to your suggestion, we have generated a new figure (Figure 3) that more effectively illustrates the spatial distribution of areas below the empirical thinning limit. This figure includes a scatter plot relating the percentage of glacier area below the empirical thinning limit (Pe ≤ 21) to the distance inland where this limit is located and a histogram showing the frequency distribution of the vulnerable regions. The figure also incorporates the clear differentiation between marine and lake terminus glaciers using separate symbols and the specific Ice Fields region (NPI and SPI).

**R3C81:** L 544: 42 km is almost the full width of the icefield! Worth pointing out.

**Response to R3C81:** Dear reviewer, we appreciate your suggestion that in the revised version of the manuscript, we have explicitly emphasized the significant inland extension of the thinning limit, considering our new results. The text added to the discussion where this feature is highlighted is as follows:

"This widespread vulnerability is particularly significant as the area below the empirical limit extends across almost the entire width of PI."

**R3C82:** L 553: I thought that Pe = 8 was the key value, but here you're writing that it's 4.85? With three significant digits? I'd think two was more appropriate given the precision of the analysis and the many approximations necessary.

**Response to R3C82:** We appreciate this comment and agree that adding three significant digits was unnecessary. Due to the new structure of the manuscript, the associated information was removed from the results section and, consequently, from the discussion section.

**R3C83:** L 578: Vulnerability is not "also" controlled by geometry. But Pe IS glacier geometry. The two points are one and the same.

**Response to R3C83:** We appreciate this important observation regarding conceptual redundancy. This section has been completely restructured in the current version of the manuscript, eliminating this redundancy.

**R3C84:** L 630: I don't quite capture the implication of this statement. Please make it more explicit.

**Response to R3C84:** In the original text, we mentioned that the difference in the surface mass balance was not included in the thickness product generated using the geodetic mass balance of Dussaillant et al. (2019). However, this no longer influences our results in the current manuscript since we excluded the force balance from our analysis.

**R3C85:** L 667-669: Be specific about the particular ways these glaciers stand out. In their distance to Pe=8? Or some other characteristic?

**Response to R3C85:** In the revised version of the manuscript, we have incorporated a more detailed and quantitative analysis of the characteristics that make these specific glaciers stand out, taking into account the new structure of the manuscript.

**R3C86:** L 680: Code should be available now at the time of review.

**Response to R3C86:** We have released the code associated with our analysis and the dataset used, including the central lines analyzed in this manuscript version. We hope this will allow a quick and user-friendly reproduction of our results and facilitate the application of the methodology in unexplored regions.